# Dropped Scheduled Task: Mitigating Negative Transfer in Multi-task Learning using Dynamic Task Dropping

**Aakarsh Malhotra**                                                    *aakarshm@iiitd.ac.in*
*Department of Computer Science*
*IIIT-Delhi, New Delhi, India 110020*

**Mayank Vatsa**                                                        *mvatsa@iitj.ac.in*
*IIT Jodhpur, Rajasthan, India 342037*

**Richa Singh**                                                         *richa@iitj.ac.in*
*IIT Jodhpur, Rajasthan, India 342037*

**Reviewed on OpenReview:** *https://openreview.net/forum?id=myjAVQrRxS*

## Abstract

In Multi-Task Learning (MTL), $K$ distinct tasks are jointly optimized. With the varying nature and complexities of tasks, few tasks might dominate learning. For other tasks, their respective performances may get compromised due to a *negative transfer* from dominant tasks. We propose a *Dropped-Scheduled Task* (DST) algorithm, which probabilistically "drops" specific tasks during joint optimization while scheduling others to reduce negative transfer. For each task, a scheduling probability is decided based on four different metrics: (i) task depth, (ii) number of ground-truth samples per task, (iii) amount of training completed, and (iv) task stagnancy. Based on the scheduling probability, specific tasks get joint computation cycles while others are "*dropped*". To demonstrate the effectiveness of the proposed DST algorithm, we perform multi-task learning on three applications and two architectures. Across unilateral (single input) and bilateral (multiple input) multi-task networks, the chosen applications are (a) face (AFLW), (b) fingerprint (IIITD MOLF, MUST, and NIST SD27), and (c) character recognition (Omniglot) applications. Experimental results show that the proposed DST algorithm has the minimum negative transfer and overall least errors across different state-of-the-art algorithms and tasks.

## 1 Introduction

Machine learning and deep learning aim to optimize an objective to achieve an end goal. Depending on the use-case scenario, the goal may vary from a regression task to a classification task. However, in the real world, there are scenarios where many objectives need to be solved together. For example, a self-driving car needs to decipher a road sign while tracking the car in front of it while simultaneously avoiding an incoming pedestrian on the road (Yang et al., 2018; Chowdhuri et al., 2019; Chang et al., 2021). Similarly, an explainable deep model can predict if an organ's X-ray/CT/MRI is infected while segmenting the disease to assist the doctor (Amyar et al., 2020; Zhang et al., 2020; Malhotra et al., 2022a). In such scenarios, algorithms need to process an input data, like a road view from a lidar sensor or an organ X-ray/CT/MRI scan, to perform multiple tasks.

In the scenarios defined above, Multi-Task Learning (MTL) (Caruana, 1997; Zhang et al., 2021a; Vandenhende et al., 2021) is an ideal choice. MTL is a joint optimization of $K$ distinct tasks. The aim is to share information across related tasks by utilizing shared representations. In a deep convolutional neural network (CNN), tasks in MTL often share network parameters or layers to design scalable and robust solutions. In MTL, these $K$ tasks can be placed for training at different stages (depth) of the network. However, at some

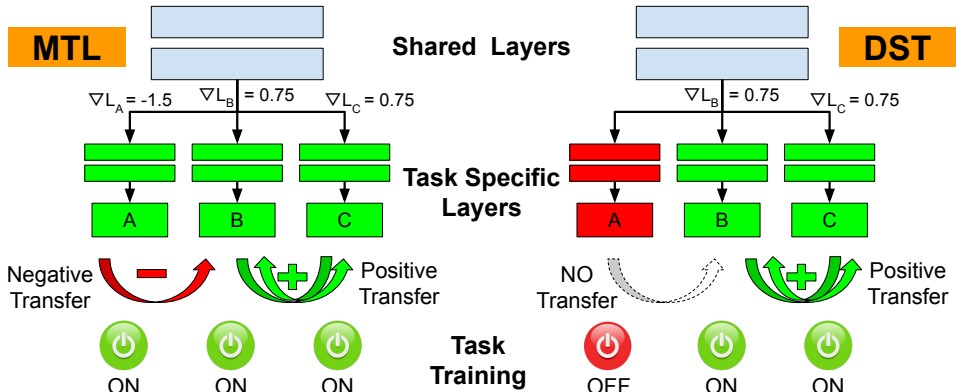

**Figure 1:** Some tasks can dominate learning in MTL, resulting in negative transfer to other tasks. The proposed Dropped Scheduled Task (DST) algorithm reduces negative transfer by dynamically selecting dominating tasks and "dropping" them. This way, other tasks get more compute cycles and learns without negative transfer.

level, different tasks share parameters and use the shared representations to perform the respective $K^{th}$ end task. The shared representations enable the generalization of the solution (Caruana, 1997; Ndirango & Lee, 2019) while saving time and cost to deploy each of the $K$ task models separately. MTL offers flexibility with tasks getting trained together at different stages (depth) and even different loss functions such as MSE or cross-entropy. With different depths, tasks requiring coarse features may be trained with a smaller network depth than more intricate tasks that require considerable network depth and more epochs. However, these variabilities in joint optimizations may lead to negative transfer (Wang et al., 2019; Zhang et al., 2021b).

## 1.1 Background and Problem Formulation

In this subsection, we provide the notations, background, and problem definition for multi-task learning and negative transfer. In MTL, the aim is to optimize for a set of $K$ tasks simultaneously on an $i^{th}$ input $X_i$. $X_i$ is an instance in a z-dimensional input dataset $X \in R^{N \times z}$ with a task-specific labels $Y_t \in Y^{N \times K}$. Note that $1 \leq |Y_t| \leq N$ translates to the fact that each task may have a different number of labeled data samples. However, each sample has at least one task with labeled ground truth. The MTL network is denoted by $f(X; \theta^k)$, where $\theta^k$ are the network parameters at $k^{th}$ epoch, consisting of shared and task-specific parameters, given as $\theta_{sh}^k$ and $\theta_t^k$ respectively. Here, $t$ refers to individual tasks, where $1 \leq t \leq K$.

MTL aims to learn the network parameters $\theta$ by simultaneously optimizing all $K$ task's losses. Hence, the net optimization for the $i^{th}$ instance becomes $\mathcal{L}_i = \sum_{t=1}^K \mathcal{L}_{t_i}$, where $\mathcal{L}_{t_i}$ is the individual loss term for the $t^{th}$ task. In a more general sense, the net optimization can be written as a linear combination of individual losses as $\mathcal{L}_i = \sum_{t=1}^K w_t \mathcal{L}_{t_i}$. The weighting term $w_t$ can be used for loss scaling or task prioritization (Gong et al., 2019). While jointly optimizing with individual task's objectives, source and target domains are learned together. During this joint learning, tasks can assist each other for improved learning and generalization. However, due to imbalanced learning between tasks, overall MTL performance or performance for a subset of task(s) can occasionally be sub-par compared to its single task learning (STL) counterpart due to a negative transfer (Ruder, 2017).

In a general sense, a negative transfer can be defined as a scenario of transfer learning where the target task demonstrates a lower performance due to knowledge transfer from a source task compared to the case where the target task is trained individually. In the context of MTL, source and target domains are trained together and interchangeably knowledge is updated. With multiple tasks optimized together, a subset of task(s) may dominate training. Here, the performance for these dominant tasks improves while the tasks outside the dominant group observe lower performance (Liu et al., 2019a), referred to as negative transfer in MTL. We describe negative transfer in MTL as a scenario where a subset of target task(s) reports sub-optimal performance compared to their respective performance when trained in the absence of source task(s). More specifically, the absence of a source task can be defined as a scenario where the target task

**Table 1:** Correlation of task-specific gradients at the shared layer on the five tasks of the AFLW database (Koestinger et al., 2011). The five tasks are: $T_1$: gender prediction, $T_2$: wearing spectacles or not prediction, $T_3$: landmark localization, $T_4$: pose estimation, and $T_5$: denoising.

|       | $T_1$ | $T_2$  | $T_3$ | $T_4$ | $T_5$  |
|-------|-------|--------|-------|-------|--------|
| $T_1$ | 1.0   | 0.012  | 0.001 | 0.005 | 0.006  |
| $T_2$ | 0.012 | 1.0    | 0.001 | 0.006 | -0.007 |
| $T_3$ | 0.001 | 0.001  | 1.0   | 0.117 | 0.044  |
| $T_4$ | 0.005 | 0.006  | 0.117 | 1.0   | 0.004  |
| $T_5$ | 0.006 | -0.007 | 0.044 | 0.004 | 1.0    |

is trained individually (single task learning: STL), or the target task is trained in an MTL setup without the source task in consideration. This can be seen in Figure 1 under the MTL setup. For instance, outliers in an already learned, well-performing, easier task can negatively affect another challenging task that is still underperforming. Alternatively, incorrect predictions from complex tasks may negatively affect correctly predicting easier tasks (Lee et al., 2018). Furthermore, some unrelated tasks might get left out during the training.

To formally persuade the problem of negative transfer, let us consider task interference on the shared parameters $\theta_{sh}^k$. During parameter updates for each sample $i$ (or minibatch), the gradients $\nabla_\theta$ flow backwards from task-specific $\theta_t^k$ to the shared parameters $\theta_{sh}^k$. The parameters $\theta_{sh}^k$ are updated by a linear combination of individual task's gradients, as given by: $\nabla_\theta^{sh} L = \sum_{t=1}^{K} \nabla_\theta L_{t_i}$. However, at these shared layers, the gradients of different tasks may interfere with opposite update directions. The disagreement in the gradient directions between the tasks may nullify the overall gradient, limiting performance for a subset of task(s). For instance, as seen for MTL in Figure 1, gradients from $T_A$ nullify gradients for $T_B$ and $T_C$ at the shared layer.

To motivate the task interference with respect to counter directions of task gradients, we showcase vanilla MTL of five tasks of the AFLW database (Koestinger et al., 2011) using the architecture shown in Figure 2. While more details follow in Section 2.2, the five tasks are: $T_1$: gender prediction, $T_2$: wearing spectacles or not prediction, $T_3$: landmark localization, $T_4$: pose estimation, and $T_5$: denoising. We obtain gradients with respect to each task for each minibatch throughout training for a randomly selected parameter in the shared encoder layer. This yields a vector of gradient values for each task. Using these vectors, we quantify the correlation of gradients across each pair of tasks (Dwivedi & Roig, 2019). As seen in Table 1, there is a limited consensus in task gradients, which supports the impact of negative transfer caused by interfering gradients.

Recent studies aim to achieve global optima and address negative transfer by manipulating the training process. For limiting negative transfer and controlling information sharing while training an MTL network, the existing approaches can be categorized under three categories: (i) task grouping, (ii) task prioritization, and (iii) curriculum learning. The upcoming subsection elaborates on each of these methods.

## 1.2 Literature Review

The commonly used approaches to optimize training process includes: (i) task grouping (Kumar & Daume III, 2012), (ii) task prioritization (Gong et al., 2019), and (iii) curriculum learning (Bengio et al., 2009). Each method aims to control the priority of the tasks to improve the overall performance. For these existing approaches, we describe each of these three hoods is below.

**Task Grouping:** In this approach, algorithms aim to group tasks by identifying their *relatedness*. Post grouping, the optimization is done by selecting a group of tasks or sub-network for better multi-task training. Research studies compute grouping by measuring affinities (Zhang & Yeung, 2014; Standley et al., 2019), cross-task consistencies (Zamir et al., 2020), or probability of concurrently simple/difficult tasks (Lu et al., 2017). Post grouping, the algorithm may give different sub-networks (Lu et al., 2017; Standley et al., 2019) or orchestrate MTL (Zhang & Yeung, 2014; Zamir et al., 2020). Instead of a single level, a few studies (Han & Zhang, 2015; Chen et al., 2018a; Zamir et al., 2018) further segregated task grouping into different levels to model complex relationships amongst tasks. Though not exactly on the lines of grouping, Maninis et al.

(2019) proposed attentive single-tasking while training a multi-task network. The idea was to select a single task to execute using a residual adapter so that the MTL network could learn focussed features relevant to the task and disregard irrelevant features. In 2020, Chen et al. (2020) suggested dropping gradients of a group of tasks based on a positive sign purity metric. The idea was to drop either negative or positive gradients from different tasks so that they do not neutralize each other during training. Similarly, Navon et al. (2022) formulated MTL as a Nash-MTL algorithm. Nash-MTL performed learning as a bargaining game, where tasks negotiate to obtain consensus for the gradient direction. Recently, Liu et al. (2022) built a dynamic task relationship based on meta-loss during training. The association of tasks is dynamically decided based on the validation loss values. The result is a dynamic task-specific weighting parameter $\lambda$ against each task's loss.

**Weight based Task Prioritization in MTL:**  In previous subsection, we defined MTL as $\mathcal{L}_i = \sum_{t=1}^{K} \mathcal{L}_{t_i}$. To prioritize tasks, research studies weigh each task individually and optimize as: $\tilde{\mathcal{L}}_i = \sum_{t=1}^{K} w_t \mathcal{L}_{t_i}$. The term $w_t$ is altered based on incompleteness/hardness of each task. Assuming task-specific loss is informative enough, studies use loss value to dynamically balance training. GradNorm (Chen et al., 2018b) altered the gradients of the network to prioritize tasks. They calculated the ratio of current loss and the starting loss for all the tasks. Each task's gradients are altered based on the calculated task-wise ratio against the expected ratio value across all tasks. LBTW (Liu et al., 2019a) uses the ratio of current loss and the starting loss to determine completeness. The term $(\text{ratio})^\alpha$ acts as a weight for each task. Other weighting strategies include studies by Sener & Koltun (2018) that finds weights leading to a Pareto-optimal solution, and Kendall et al. (2018) that uses homoscedastic uncertainty for task weighing. Similarly, Liu et al. (2019b) introduced Dynamic Weight Average (DWA) to calculate relative descending rate. DWA is then used to weigh individual tasks. Liebel & Körner (2018) added task-wise weights into learnable parameters. They constrained these weights with a regularization term to enforce non-trivial solutions. Other than loss values, studies (Guo et al., 2018; Jean et al., 2019) also utilize train or validation set performance to determine completeness or hardness of the task. In some scenarios, weights are dynamically modified to achieve performance closer to pre-computed Single Task Learning (STL).

**Curriculum Learning:**  In 1993, Elman (1993) discussed the concept of *starting small* to improve the training process for multiple task subsets. However, the term "*curriculum learning*" was coined by Bengio et al. (2009). The idea was to learn easier examples/samples followed by introducing more challenging cases. Results showed a speed up in the training process by finding good local minima for non-convex learning. Lee & Grauman (2011) use the idea to sequentially discover difficult categories using the earlier discovered easier ones without supervision. Pentina et al. (2015) performed MTL by sequentially learning individual tasks. They regularized parameters of each introduced task to be similar to just previously learned tasks. The idea was that consecutive related tasks in the order of difficulty should have similar parameter representation. Recently, Graves et al. (2017) utilized two signals as a reward to tune curriculum learning. These signals were the amount of increase in accuracy and increase in model complexity.

Despite advancements in task grouping, task prioritization, and curriculum learning, negative transfer remains a predominant issue in MTL. The limitations of existing approaches are: (i) they seldom consider network depth while training that can cause a negative transfer. Further exacerbated negative transfer may be caused by a task-wise varying count of ground-truth annotated training samples (Ge et al., 2014; Wu et al., 2020), (ii) the weighted approaches may leave some tasks in a stagnant condition, and lastly, (iii) a negligible weight to early completed tasks can result in their catastrophic forgetting. Hence, this research aims to discover potential issues that can cause a negative transfer and address them using the proposed optimal training approach.

**Research Contributions:** In this research, we propose Dropped-Scheduled Task (DST) to address negative transfer by dynamically selecting dominating tasks and "dropping" them. Our approach selects four factors that cause negative transfer and resolves them using the corresponding four metrics. These include (i) network depth, (ii) ground-truth availability count, and current loss values to determine (iii) task-wise learning completeness, and lastly, (iv) task-wise stagnation. Using these parameters, the DST occasionally "drops" a subset of tasks by a task-specific activation probability. The term "dropping" refers to not considering a subset of tasks during joint optimization, i.e., preventing error propagation through the dropped task(s). As shown in Figure 1, holding/dropping quick learners gives a fair chance to complex tasks, and all

the tasks are learned without negative transfer. Dropping can also prevent dominance/overfitting (Wan et al., 2013; Srivastava et al., 2014; Ghiasi et al., 2018) of the completed tasks. Along these lines, a recent work by Sun et al. (2021) suggested performing one task at a time during MTL optimization. However, different combinations of tasks learning together can act as a regularization method and result in the generalization of the solution (Caruana, 1997; Ndirango & Lee, 2019). The related tasks may further assist other data scarce tasks and assist in their learning as well (Kapidis et al., 2021). The contributions of the DST algorithm are as follows:

- The DST algorithm considers negative transfer caused due to task-wise varying network depths and limited samples for some tasks. Due to network depth as a parameter, DST is shown to work in unilateral and bilateral models.

- The DST algorithm computes completeness and stagnation of the task, allowing only a subset of tasks to remain active. Further, DST allows prolonged stagnated earlier finished tasks to be occasionally active in later training stages, reducing catastrophic forgetting.

- Due to task *dropping*, $2^K$ combinations from $K$ tasks are possible. Different combinations during training act as implicit regularization to move tasks out of local minima.

- For experimental results, three applications are chosen related to faces, (latent) fingerprints, and character recognition. The applications are chosen considering: (i) their diverse nature and complexities (classification, image-to-image translation, regression, contrastive loss, segmentation), (ii) different output depths (encoder/FC/decoder) and architecture (unilateral/bilateral), (iii) varying difficulty of similar tasks (segmentation for fingerprints easier than latent fingerprints), and (iv) large number of tasks (Omniglot).

## 2 Dropped-Scheduled Task (DST)

Negative transfer in MTL can be due to a variety of factors. The proposed DST algorithm[1] is based on four important factors that can negatively affect learning in MTL. Based on the four factors, the proposed DST algorithm devises four metrics, based on which a task-specific activation probability is computed. Consequently, during training, the DST algorithm occasionally "drops" a subset of tasks based on the computed task-specific activation probability.

The four factors are network depth, ground-truth sample count, task incompleteness, and task stagnation. For each $t^{th}$ task in MTL, these four factors are quantified by a respective metric to quantify the drop rate. For $t^{th}$ task, $\mathcal{P}_{(d,t)}$ quantifies drop rate based on task depth and $\mathcal{P}_{(c,t)}$ quantifies drop rate based on training sample count. Additionally, at $k^{th}$ epoch, $\mathcal{P}_{(u,k,t)}$ and $\mathcal{P}_{(r,k,t)}$ quantifies drop rate based on task incompleteness and stagnancy, respectively. These individual metrics tell us how a certain task is overpowering or underpowered. These metrics are combined to define a task-wise activation probability $P_{(k,t)}$, where $P_{(k,t)}$ ranges in $[0,1]$. $P_{(k,t)}$ tells what are the chances of $t^{th}$ task to remain active on $k^{th}$ epoch. The following section explains the proposed DST algorithm and the four metrics. It is followed by details of the chosen tasks and respective architectures.

### 2.1 Dropping Mechanism to Hold Overpowering Tasks

In the proposed DST algorithm, a task $t$ can remain active at $k^{th}$ epoch by a probability $P_{(k,t)}$. Otherwise, the tasks are *dropped* by a probability $(1 - P_{(k,t)})$. $P_{(k,t)}$ is a weighted combination of five different metrics (one being a regularizer). The following subsections explain how the factors like network depth, ground-truth sample count, task incompleteness, and task stagnation affect MTL. Further, we explain how these adverse factors can be translated into individual metrics, which eventually are combined to get task activation probability $P_{(k,t)}$.

---

[1]The source code is available at: https://github.com/aakarshmalhotra/DST.git.

### 2.1.1 Metric based on Network Depth

He et al. (2016) highlighted that increasing the layers in a deep network may alleviate the problem of vanishing gradients. Vanishing gradients refer to a scenario during backpropagation where the gradient diminishes as the network depth increases. Few studies utilizing multi-task networks have addressed the concerns using a residual connection (Subramanian et al., 2018; Ding et al., 2019; Li et al., 2020) or some additional loss terms (Qiu et al., 2022). These approaches work well when all tasks are placed at the same depth. In an MTL setup with tasks of different depths, the effect of gradients can differ on the initial layers. For a shallower task, the gradient would dominate the initial layers. On the other hand, gradients would relatively vanish for deeper tasks.

To give an equal chance, one way is to increase the gradient. GradNorm (Chen et al., 2018b) showed a mechanism to alter gradients for faster training. Considering task-wise network depths, our study gives more computation cycles (keeping tasks active) for deeper tasks. It reduces the dominance of shallower tasks. Based on network depth $d$ for task $t$, the metric $\mathcal{P}_{(d,t)}$ is:

$$\mathcal{P}_{(d,t)} = \frac{d_t}{\max\limits_{1 \leq t \leq K} (d_t)}. \tag{1}$$

A higher value of $\mathcal{P}_{(d,t)}$ represents that the task considered is deeper and increases the chances of computation cycles.

### 2.1.2 Metric based on Training Sample Count

While training a deep network, the model may overfit if the training samples are low and trained for a long duration (higher number of epochs). On the contrary, the network may underfit if it is trained for lesser epochs with larger number of training samples. For MTL, not all tasks may have an equal number of ground-truth labels. The task with fewer annotated samples might benefit (positive transfer) from the large corpus of samples provided by another task with more labeled samples. However, a target task with more labeled samples might undergo a negative transfer due to a source task with fewer instances (Wu et al., 2020). This can be seen from Lee et al. (2016) and our experiments, where a highly confident task faces a negative transfer due to a low-confident task trained with fewer training samples. Hence, each task requires computing cycles proportional to the number of labeled instances.

We propose to consider task wise ground-truth count to decide computation cycles for each task. It ensures a reduction in computation cycles for tasks with fewer annotated training samples by dropping them more often. On the other hand, it increases computation cycles for the task with more annotated training samples. The metric $\mathcal{P}_{(c,t)}$ is given as:

$$\mathcal{P}_{(c,t)} = \frac{c_t}{\max\limits_{1 \leq t \leq K} (c_t)}. \tag{2}$$

Here, $\mathcal{P}_{(c,t)}$ is the ratio of the ground-truth count $c_t$ of the considered $t^{th}$ task divided by the maximum ground-truth availability count of all tasks. A higher value of $\mathcal{P}_{(c,t)}$ represents that the task has more labeled instances, resulting into task having more computation cycles.

### 2.1.3 Metric based on Task Incompleteness

To dynamically perform task scheduling, the proposed algorithm assumes that task-specific losses are descriptive for task balancing. A ratio of each task's current loss to its initial loss shows how much the task has learned. The expected value of this ratio across tasks denotes the average completion rate. Using the expected ratio, we can establish a relative completeness/incompleteness for each task.

Let the value of loss for task $t$ at the $k^{th}$ epoch be $V_{(k,t)}$. The initial loss value after the $1^{st}$ epoch is represented as: $V_{(1,t)}$. Hence, the amount of "incompleteness" during training for task $t$ at the $k^{th}$ epoch can be defined as:

$$I_{(k,t)} = \frac{V_{(k,t)}}{V_{(1,t)}}. \tag{3}$$

For a task whose learning is incomplete with a negligible decrease in loss, the value for $I_{(k,t)}$ would be close to 1. On the contrary, a task that has completely learned would have a small value towards 0. Further, it could be a case where a task has digressed and has its $V_{(k,t)} \geq V_{(1,t)}$. This would result in $I_{(k,t)} \geq 1$. Using $I_{(k,t)}$, the relative incompleteness of each task $t$ can be found as:

$$\mathcal{P}_{(u,k,t)} = \min\left(1, \frac{I_{(k,t)}}{E(I_{(k)})}\right). \tag{4}$$

Here, $E(I_{(k)})$ is the expected value of $I_{(k,t)}$ across all tasks at $k^{th}$ epoch. The variable $u$ in $\mathcal{P}_{(u,k,t)}$ is an alias for incompleteness measure, as defined by $I_{(k,t)}$. $\mathcal{P}_{(u,k,t)}$ penalizes faster learning tasks but does not reward a slow task as the value is upper bound to 1. It makes all tasks that are "uncompleted" and slower than estimated $E(I_{(k)})$ to obtain the highest activation chance while reducing the compute cycles of faster tasks.

### 2.1.4 Metric based on Task being Stagnant

While training, some tasks may become stagnant in an MTL setting. Stagnancy is defined as a phase where the rate of fall of training loss is negligible. An easy, quick learning task may become stagnant due to training completion. Such quick learning tasks should be refrained from further training to avoid overfitting. On the other hand, a few tasks may get left out due to not getting priority in MTL. Such tasks require more explicit emphasis with more compute cycles. Hence, the metric presented in this subsection aims to empower incomplete stagnant tasks.

We now devise a metric to quantify incompleteness and stagnance together. For each task, stagnancy can be computed by monitoring the decline in loss values over a few epochs. For the same, we calculate the local rate of change of losses using the loss value in the current and previous epoch. The local rate of change of loss $R_{(k,t)}$ for $t^{th}$ task at $k^{th}$ epoch can be defined as:

$$R_{(k,t)} = \frac{1}{\mathcal{P}_{(u,k,t)}} \times \frac{V_{(k,t)} - V_{(k-1,t)}}{V_{(k-1,t)}} \qquad \forall k \quad k \geq 2. \tag{5}$$

Here, $V_{(k,t)}$ represents the loss value for task $t$ at the $k^{th}$ epoch. The term $\mathcal{P}_{(u,k,t)}$ is taken from Eq. 4 to encode task incompleteness. The multiplication by $\frac{1}{\mathcal{P}_{(u,k,t)}}$ ensures that for two equally stagnant tasks, priority is given to the relatively incomplete one. On the other hand, the term $\frac{V_{(k,t)} - V_{(k-1,t)}}{V_{(k-1,t)}}$ is responsible to encode stagnancy across two simultaneous epochs.

Next, we need to compute if an incomplete task is in a stagnant phase rather than being stagnant in just two simultaneous epochs. Using the local rate of change $R_{(k,t)}$, we compute the global exponential moving average as:

$$R'_{(k,t)} = \begin{cases} R_{(k,t)} & \text{if } k = 2; \\ \beta R_{(k,t)} + (1-\beta)R'_{(k-1,t)} & \text{if } k \geq 3. \end{cases} \tag{6}$$

Here, $\beta \in [0, 1]$ is the discount factor. Larger values of $\beta$ prioritizes more recent rate of change. $R'_{(k,t)}$ is assigned value of $R'_{(k-1,t)}$ if $R_{(k,t)} \leq 0$. Here, $R'_{(k,t)} \in (0, \infty)$, where a value close to 0 indicates an incomplete and stagnant task. The metric based on loss stagnancy is defined as:

$$\mathcal{P}_{(r,k,t)} = \begin{cases} 1 & \text{if } k = 1; \\ \min\left(1, \frac{E(R'_{(k)})}{R'_{(k,t)}}\right) & \text{if } k \geq 2. \end{cases} \tag{7}$$

Here, $E(R'_{(k)})$ denotes the expected value of $R'_{(k)}$ across all tasks at the $k^{th}$ epoch. The variable $r$ in $\mathcal{P}_{(r,k,t)}$ is an alias for stagnancy measure, as defined by $R'_{(k,t)}$. Stagnant and incomplete tasks obtain a value of $\mathcal{P}_{(r,k,t)}$ closer to 1.

### 2.1.5   Regularization

The above four metrics control chances of task activation based on network depth, ground-truth sample count, task incompleteness, and task stagnation. To prevent a task to completely remain OFF, resulting in catastrophic forgetting, each task needs to get some chance to be active. To do so, a regularization metric is given as $\mathcal{P}_{(b,t)} = 1$.

### 2.1.6   Final Metric for Dropping and Scheduling

Using the five metrics defined above, the final probability to schedule a particular task $t$ at $k^{th}$ epoch is:

$$P_{(k,t)} = \lambda_d \mathcal{P}_{(d,t)} + \lambda_c \mathcal{P}_{(c,t)} + \lambda_u \mathcal{P}_{(u,k,t)} + \lambda_r \mathcal{P}_{(r,k,t)} + \lambda_b \mathcal{P}_{(b,t)}. \tag{8}$$

Here, $\lambda$ are non-negative weights given to individual metrics such that $\sum \lambda_i = 1$. These $\lambda_i$ can be altered to address specific variations of data, network, and learning in MTL setup. For our experiments, details and ablation around $\lambda_i$ are shown in Section 3.2 and 4.4 respectively. Using $P_{(k,t)}$ defined above, $G_{(k,t)}$ is sampled as:

$$G_{(k,t)} \sim \text{Bernoulli}(P_{(k,t)}). \tag{9}$$

$G_{(k,:)}$ is a 1×K vector of independent Bernoulli random variables for K tasks. For a task $t$ such that $1 \leq t \leq K$, the $t^{th}$ value in the vector denotes an ON/OFF bit for the $t^{th}$ task at $k^{th}$ epoch (during training).

$$\tilde{\mathcal{L}}_{(k,t)} = G_{(k,t)} \odot \mathcal{L}_{(k,t)}. \tag{10}$$

Here, $\odot$ denotes element-wise multiplication. The vector $G_{(k,t)}$ is sampled and multiplied element-wise with individual task losses. The multiplication accounts for sampling and scheduling tasks from all sets of tasks at each epoch.

The proposed DST algorithm schedules and gives compute cycles to one or more tasks. The advantages of the proposed DST algorithm is as follows:

- Dynamic scheduling according to task-wise loss values.

- With K tasks being ON or OFF, each combination from $2^K$ combinations is possible. This acts as an implicit regularizer that may take a particular task out of its local minima due to varying combinations of active tasks.

- For two stagnant tasks, DST provides more compute cycles to the relatively incomplete task.

- The DST method considers network depth and ground-truth training sample count for each task.

- The occasional OFF may take easier tasks backward and again take them to their minima when they are ON. This prevents their overfitting. In later stages of training, the learning allows the network to gradually switch ON the completed tasks at the very end (since they have been stagnant over time) avoiding their catastrophic forgetting.

Next, we discuss the applications considered in this study and their corresponding architectures. We show MTL on faces, (latent) fingerprints, and character recognition. The face and character recognition use the hard parameter sharing MTL approach. On the other hand, (latent) fingerprint recognition uses a soft parameter sharing MTL approach. We refer to the face application architecture as unilateral MTL. In unilateral MTL, an image input is fed, and the tasks segregate out after a few shared layers. The same idea

is followed in the application of character recognition. (Latent) fingerprints use bilateral MTL for various tasks. In bilateral MTL, two images are taken as input and processed with soft parameter sharing. We elaborate on the details of these architectures in the following two subsections.

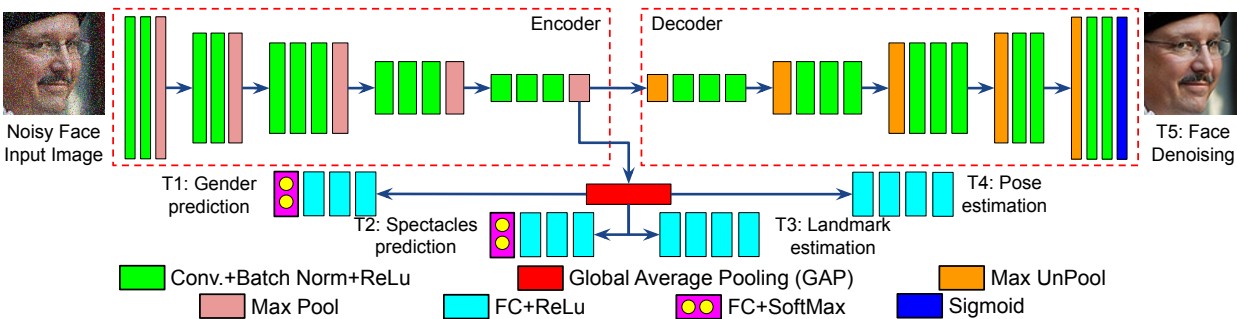

**Figure 2:** The Encoder-Decoder MTL architecture for face applications. The five tasks are: (i) gender prediction, (ii) spectacles prediction, (iii) 21 point (x,y) landmark prediction, (iv) pose estimation, and (v) image denoising.

## 2.2 Unilateral MTL: Face Attribute Analysis

For application of attribute analysis in faces, deep MTL has been explored (Han et al., 2017; Ranjan et al., 2017). Unilateral MTL takes a face image as input and performs MTL. For MTL using face images, the selected tasks are (i) gender, (ii) wearing spectacles or not, (iii) landmarks, (iv) pose estimation, and (v) image denoising. Let the $i^{th}$ input to the network shown in Figure 2 be a noisy face image $X'_{F_i}$. Let $f_1$ represent the sub-network that predicts the gender probability $P(g_i|X'_{F_i})$ of the $i^{th}$ noisy input $X'_{F_i}$. Then, the loss function for the first task $T_{1F_i}$ of gender prediction is:

$$T_{1F_i} = -g_i log(P(g_i|X'_{F_i})) - (1-g_i)log(P(g_i|X'_{F_i})), \tag{11}$$

where, $g_i \in \{0,1\}$. Similarly, the loss for the second task of predicting if the face is "wearing spectacles/or not" can be given as:

$$T_{2F_i} = -s_i log(P(s_i|X'_{F_i})) - (1-s_i)log(P(s_i|X'_{F_i})), \tag{12}$$

where, $s_i \in \{0,1\}$. The above two tasks $T_{1F_i}$ and $T_{2F_i}$ are binary classification tasks. The other three tasks are regression tasks.

The task $T_{3F_i}$ predicts 21 distinct facial landmarks, each represented as a $\{x,y\}$ tuple. Hence, for each input $X'_{F_i}$, 42 values are predicted using a sub-network $f_3$ as:

$$\hat{Y_{l_j}} = f_3(X'_{F_i}) \qquad \forall j \quad 1 \le j \le 42. \tag{13}$$

These values are optimized using Mean Squared Error (MSE) for the third task $T_{3F_i}$, given as:

$$T_{3F_i} = \frac{1}{\sum p_j} \sum_{j=1}^{42} (p_j(Y_{l_j} - \hat{Y_{l_j}}))^2. \tag{14}$$

Here, $Y_{l_j}$ are the ground truth landmarks and $\hat{Y_{l_j}}$ are the predicted landmarks. Due to facial pose or a partial face, some landmarks may be missing. To ensure such cases keep the loss unaffected, the error is multiplied with $p_j$ (1 if the $j^{th}$ landmark is present, else 0). The error is normalized by the total number of landmarks present ($\sum p_j$).

The fourth task $T_{4F_i}$ predicts pose angular values. The head pose is defined as three rotation angles: yaw, pitch, and roll. Unlike landmarks, each of the three angles is present. Hence, the MSE for the fourth task $T_{4F_i}$, given as:

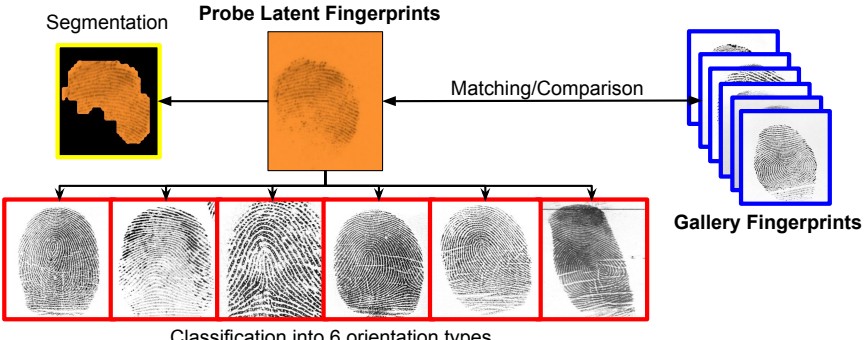

**Figure 3:** Different task associated with latent fingerprint analysis.

$$T_{4F_i} = \frac{1}{3} \sum_{j=1}^{3} (Y_{a_j} - \hat{Y_{a_j}})^2. \tag{15}$$

Here, $Y_{a_j}$ is the ground-truth while $\hat{Y_{a_j}}$ is the predicted angle. The error is normalized by total pose types, i.e., 3.

Lastly, $T_{5F_i}$ is a regression task to denoise the noisy input $X'_{F_i}$. Denoising is achieved using the sub-network $f_5$. For each pixel $j$, $T_{5F_i}$ is given as:

$$T_{5F_i} = \frac{1}{n} \sum_{j=1}^{n} (X_{F_{ij}} - f_5(X'_{F_{ij}}))^2, \tag{16}$$

where $n$ is the total pixel count and $X_{F_i}$ is the clean face image.

**Overall Loss Function:** For images with unavailable ground truth labels for a task, respective sub-networks may not be active during training. Further, the proposed DST algorithm may "switch OFF" (or drop) a task. Hence, the final loss $\mathcal{L}_i$ is computed as:

$$\tilde{\mathcal{L}}_i = \sum_{t=1}^{5} G_t w_t Z_{ti} T_{tF_i}. \tag{17}$$

Here, $Z_{ti}$ are the switches to denote the presence of the ground-truth label for $i^{th}$ image of the $t^{th}$ task. Further, $G_t$ is a gate to control training for $t^{th}$ task by the DST algorithm. The values of these switches (gates) are either 0 or 1. $w_t$ is a static weight normalization for each task by scaling the loss values to the same expected starting loss value. The term $w_t$ is optional, and the proposed DST algorithm also works for varied starting loss values across tasks.

### 2.3 Bilateral MTL: (Latent) Fingerprint Analysis

Bilateral MTL inputs a pair of images to simultaneously process them for performing joint MTL. We chose latent fingerprint analysis as it requires important tasks (Malhotra et al., 2018) of recognition, orientation classification, and segmentation. These tasks are explained in Figure 3. The selected tasks are (i) fingerprint orientation estimation, (ii) fingerprint segmentation, (iii) latent fingerprint orientation estimation, (iv) latent fingerprint segmentation, and (v) pairwise matching. The orientation of the (latent) fingerprint is a pattern in which the ridges flow. They are classified into six categories, as shown in Figure 10 in appendix. As shown in Figure 4, let the $i^{th}$ input to the Siamese network be a pair of images, fingerprint $X_{P_i}$ and latent fingerprint $X_{L_i}$.

Let $f'_1$ be the sub-network that predicts the fingerprint orientation probability $P(o_i|X_{P_i})$ in one of the pre-defined six orientations. Then, the first task can be defined as:

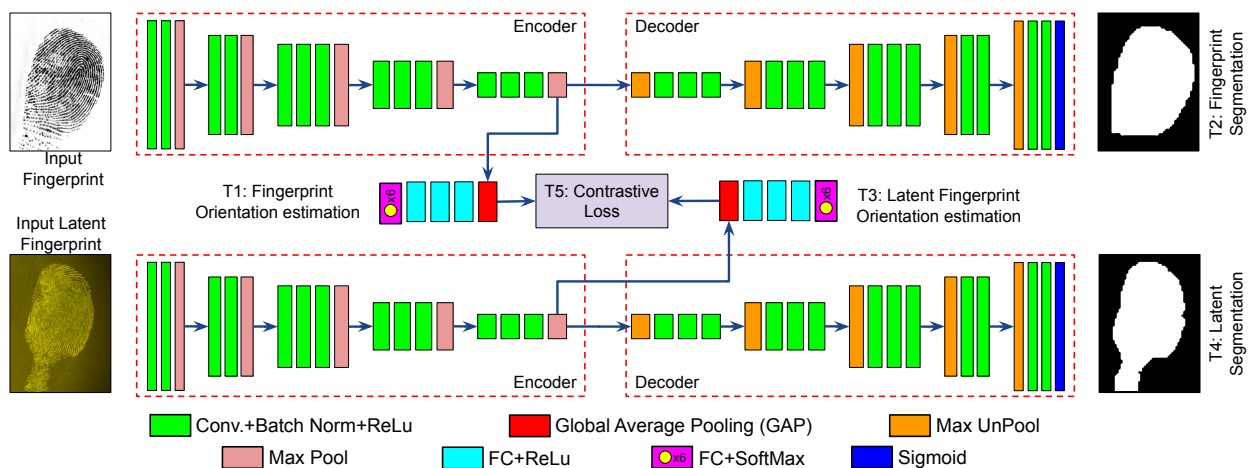

**Figure 4:** The MTL architecture for fingerprint applications. The tasks are (i) fingerprint orientation estimation, (ii) Fingerprint segmentation, (iii) latent fingerprint orientation estimation, (iv) Latent fingerprint segmentation, and (v) matching.

$$T_{4F_i} = -\sum_{i=1}^{6} o_i log(P(o_i|X_{P_i})). \tag{18}$$

Similarly, using $P(o_i|X_{L_i})$, loss function for latent fingerprint orientation prediction $(T_{3L_i})$ can also be defined.

The second and fourth tasks are semantic segmentation of fingerprint $X_{P_i}$ and latent fingerprint $X_{L_i}$ respectively. For fingerprint segmentation, let ground truth segmentation mask be $S_{P_i}$ while the predicted mask be $\widehat{S}_{P_i}$. Then, the second task $T_{2P_i}$ is defined as:

$$T_{2P_i} = -\sum_{x,y} \left[ \mathbf{S}_{\mathbf{P}_i}(x,y) \log(\widehat{\mathbf{S}}_{\mathbf{P}_i}(x,y)) + (1 - \mathbf{S}_{P_i}) \log(1 - \widehat{\mathbf{S}}_{P_i}(x,y)) \right]. \tag{19}$$

Similarly, $T_{4L_i}$ can be defined for latent fingerprint segmentation. Each of the two segmentation tasks, $T_{2P_i}$ and $T_{4L_i}$, operate at pixel level $(x, y)$. The classification for each pixel is a 2-class problem, where the masks are predicted using sub-networks $f_2'$ and $f_4'$ (with identical weights). Lastly, $T_{5_i}$ calculates pairwise contrastive loss using Euclidean distance $D_i$ between encoder representations of $X_{P_i}$ and $X_{L_i}$ as:

$$T_{5_i} = (Y_i)\frac{1}{2}(D_i)^2 + (1 - Y_i)\frac{1}{2}\max(0, m - D_i)^2. \tag{20}$$

Here, $m$ denotes the margin parameter. $Y_i = 1$ for a genuine pair, while $Y_i = 0$ for an imposter pair.

**Overall Loss Function:** Taking into account the missing ground truth labels $(Z)$, proposed task gating $(G)$, and loss value normalization $(w)$, the net loss $\tilde{\mathcal{L}'}_i$ is computed as:

$$\tilde{\mathcal{L}'}_i = G_1 w_1 Z_{1i} T_{1P_i} + G_2 w_2 Z_{2i} T_{2P_i} + G_3 w_3 Z_{3i} T_{3L_i} + G_4 w_4 Z_{4i} T_{4L_i} + G_5 w_5 Z_{5i} T_{5_i}. \tag{21}$$

## 3 Experimental Details

### 3.1 Databases and Protocols

In this study, two kinds of experiments are performed. For the first experiment with **unilateral MTL**, we have taken two case studies with two different datasets. The first case study is on faces, while the second is

**Table 2:** Train-test split for the AFLW database, with unequal ground-truth label count selected for each task.

| Task No. | Task | Task Depth | Training | | Testing (Count) |
|---|---|---|---|---|---|
| | | | Count | (%) | |
| 1 | Gender | 17 | 8434 | 50 | |
| 2 | Spectacles | 17 | 12731 | 75 | |
| 3 | Landmark | 17 | 12727 | 75 | 7384 |
| 4 | Pose angle | 17 | 6692 | 40 | |
| 5 | Denoising | 26 | 17000 | 100 | |

**Table 3:** Consolidated fingerprint database used for experiments. The labels counts are as per the source availability.

| Task No. | Task | Task Depth | Training | | Testing (Count) |
|---|---|---|---|---|---|
| | | | Count | (%) | |
| 1 | FP Orientation | 17 | 33340 | 79.12 | |
| 2 | FP Segmentation | 26 | 33340 | 79.12 | |
| 3 | LFP Orientation | 17 | 32308 | 76.76 | 13577 |
| 4 | LFP Segmentation | 26 | 9756 | 23.15 | |
| 5 | Matching | 13 | 42140 | 100 | |

on character recognition, with experiments performed on the AFLW and OmniGlot datasets, respectively. The second experiment of **Bilateral MTL** is performed on latent fingerprints. A summary of the datasets is given below.

**1. Unilateral MTL:** For **faces**, we use the **AFLW dataset (Koestinger et al., 2011)** having 24,384 usable faces. We use 70% (17,000) for training and 30% (7,384) for testing. The cropped face images are deteriorated by adding speckle noise to 15% of pixels. Table 2 shows that a varying proportion of training labels are selected for each task. For the testing set, no tags are discarded.

**Omniglot database (et al., 2015)** consists of 50 handwritten alphabets, each of which has its respective **character recognition** tasks. Omniglot is a relevant database for the application of MTL since a large number of classes can illustrate the performance of the proposed method as a function of the number of classes. Furthermore, the tasks are related as knowledge of the alphabets can help classes to learn from each other. The dataset consists of grayscale images of resolution 105×105, with 20 instances for each character. We follow the standard MTL protocol as suggested in the literature (Liang et al., 2018; Meyerson & Miikkulainen, 2018; Prellberg & Kramer, 2020). A fixed random training and testing split is created by selecting a split of 20% as the test set from each alphabet, with the remaining 80% considered training. The architecture and experimental details are highlighted in Section 4.3.

**2. Bilateral MTL:** We use a consolidated set from three fingerprint datasets: **NIST SD 27 (NIST-SD-27, 2000), IIITD MOLF (Sankaran et al., 2015), and MUST (Malhotra et al., 2022b)**, with numbers shown in the Table 3. The pairs for the fifth task are considered in a subject-disjoint split. Details for each of the database is listed below:

**NIST SD 27 (NIST-SD-27, 2000)** has 258 latent fingerprints with respective inked fingerprint, resulting in 258 genuine pairs. An equal number of imposter pairs are added, making a total of 516 pairs. In NIST SD 27, there are segmentation masks for latent impressions. Further, each fingerprint and latent fingerprint are manually annotated for orientation. All of the 516 pairs are included in training set, with labels available for only four tasks (T1, T3, T4, and T5).

**IIITD MOLF (Sankaran et al., 2015)** has 4,400 latent fingerprints (DB4). A genuine and an imposter pair with a fingerprint is formed for each latent fingerprint, totalling 8,800 pairs. Each pair has only the label for the fifth task (match/non-match) and all 8,800 pairs are included in training set.

**IIITD MUST Latent Fingerprint database (Malhotra et al., 2022b)**: As per the defined protocol for the MUST database, there is a subject disjoint train-test split. Using the train identities, 32,824 train pairs are formed (with an equal number of genuine and imposter pairs). The knowledge of genuine and imposter pairs affirms the match/non-match information for the fifth task. Finally, the MUST database's

test set is used for the testing procedure for the multi-task network. The test set has 13,577 query latent fingerprints, which are matched against all the test gallery fingerprints. The ground-truth labels for the testing set have information for all pairs for orientation (T1, T3) and matching (T5). However, for a few pairs, the segmentation masks are missing.

## 3.2 Implementation Details

The pseudo-code for task-wise sampling of activation bits using the DST algorithm is in the appendix. The models are trained on a machine with Intel Core i7 with 128 GB RAM and NVIDIA GeForce RTX 2080Ti GPU in a PyTorch implementation. Learning rate is set at $5 \times 10^{-5}$ with Adam optimizer. The encoder in MTL is initialized with VGG16 (Simonyan & Zisserman, 2014) pre-trained weights of the ImageNet dataset. During training, error backpropagation and weight update are performed from the second epoch to obtain initial loss estimates. $\beta$ in Eq. 6 is set as 0.1 to ensure a slower change for the 'stagnant' parameter. Further, all $\lambda_i$ in Eq. 8 are kept as 0.2 to provide equal weight to each metric. For face MTL, the input is a cropped face image of size 224×224×3. The model is trained for 100 epochs with a batch size of 16. For fingerprint MTL, the input is a fingerprint-latent fingerprint image pair at resolution 340×280×3. The model is trained for 75 epochs with a batch size of 4.

## 4 Results and Analysis

This section shows the efficacy of the proposed DST algorithm. The chosen applications are: (a) multi-task face (AFLW (Koestinger et al., 2011)), (b) multi-task fingerprint (IIITD MOLF (Sankaran et al., 2015), MUST (Malhotra et al., 2022b), and NIST SD27 (NIST-SD-27, 2000)), and (c) multi-task character recognition (Omniglot (et al., 2015)). Subsequently, in Section 4.4, we present an ablation study to study the effects of each selected metric, $\lambda_i$, $w_t$, network depths and initialization, and varying sample counts.

As seen in Table 4 to, Table 5, and Table 7, the proposed DST algorithm reports minimum average errors across all the three applications (face: 10.75%, fingerprint: 18.87%, character: 10.41%). We compare the results against STL, MTL and random task drop. Furthermore, the results are also compared against other task prioritization or scheduling algorithms (Lee et al., 2018; Chen et al., 2018b; Guo et al., 2018; Jean et al., 2019; Liu et al., 2019a; Maninis et al., 2019; Chen et al., 2020; Liu et al., 2022) with varying necessary parameters. Furthermore, the comparative results show that $\alpha = 0.5$ are the desired parameters for GradNorm (Chen et al., 2018b) and LBTW (Liu et al., 2019a). After the best performanc by the proposed DST, we also observe that LBTW (Liu et al., 2019a), Gradient Drop (Chen et al., 2020), and Auto-$\lambda$ (Liu et al., 2022) result in competitive performance across the three applications. We derive information from the train set for all algorithms that require task-wise performance/loss values to weigh/schedule tasks.

## 4.1 Unilateral MTL: Faces

When MTL is performed for different tasks with input as face images, the results are shown in Table 4. Visually, prediction outputs for denoising and landmark prediction are shown in Figure 8 and 9 of the appendix, and briefly in Figure 7 on this main paper.

We observe that the regression tasks (T3, T4, and T5) undergo negative transfer compared to STL. On the contrary, gender classification improves by 1.15% over corresponding STL. The gender prediction task has 40% labeled samples. Hence, learning from other tasks helps in a performance gain for gender task. By allowing a negligible drop in gender classification performance, DST improves performance for all the other four tasks and reports the least average error. For landmark localization, the DST algorithm provides a 1.3% lower error than the STL. This can be seen from loss and task activation plots in Figure 6. The relatively more computation cycles for landmark localization with the occasional absence of others help task 3 move towards its minima. Concurrently, the occasional ON/OFF of tasks 1, 2, and 4 limits trained tasks from overfitting. Once trained towards the end, the DST algorithm also reduces the activation probability of T3.

Further, to intuitively illustrate the effect of DST, we study two tasks using their T-Sne plot in Figure 5. When tasks of gender classification (T1) and "wearing spectacles" (T2) are trained as a vanilla MTL, we get

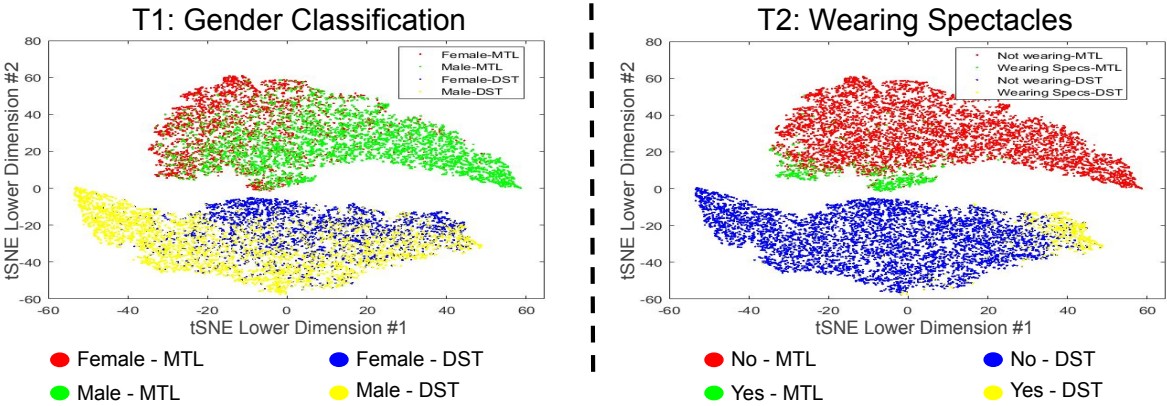

**Figure 5:** t-SNE plot for gender classification (T1) and "wearing spectacles" (T2) in a vanilla MTL & and with the proposed DST algorithm. The axis represent a 2D projection of high dimensional features (encoder representation of size $1 \times 512$), as obtained by the t-SNE algorithm. Notice the two green clusters in the right t-SNE plot. As the gender task dominates, we see that it overpowers the T2 of "wearing spectacles", resulting in three clusters for T2: (i) not wearing spectacles (red), (ii) females with spectacles (left green), and (iii) males with spectacles (right green). Without gender classification, we would have seen just the two groups of clusters. Using the proposed DST algorithm, we observe the lower blue-yellow representation points. While gender classes are separated similarly (left t-SNE plot), the wearing vs. non-wearing spectacles is now just two clusters (right t-SNE plot).

the upper kidney-like representation cloud (green-red points). As gender task dominates, it overpowers the "wearing spectacles" task. In the T2 plot on the right, green points illustrate wearing spectacles using the vanilla MTL algorithm, while red is not wearing spectacles. Hence, when the person is "wearing spectacles", two clusters are formed (one for men with spectacles and the other for women with spectacles). The consequence is that for spectacles, there are three clusters of data, (i) not wearing spectacles, (ii) females with spectacles, and (iii) males with spectacles. Without gender classification, we would have seen just the two groups of clusters. Using the proposed DST algorithm, $P_{(c,t)}$ metric reduces the dominance of T1 and T2 gets more chance. T2 then makes relevant weight updates with limited influence of T1, allowing T2 to separate the "wearing spectacles" class as one cluster. Hence, we now observe in Figure 5 the lower blue-yellow representation points. While gender classes are separated similarly, the wearing vs. non-wearing spectacles is now just two clusters.

Compared to other task prioritization algorithms, the proposed DST algorithm has the best overall performance and the least negative transfer. However, for $\alpha \geq 1$, GradNorm and LBTW drift away from the optimal solution and fail to work on the easier tasks (Gender and Spectacles). DTP (Guo et al., 2018) operates on key performance indicators. On the other hand, Deep Asymmetric MTL by Lee et al. (2018) aims to focus more on reliable predictors from easy tasks. Hence, the results for Deep Asymmetric MTL is much improved upon easier tasks of gender and spectacles. However, overall, the average error deteriorates to 11.52, mainly due to lower performance on the difficult task of landmark localization. Even though a task may have saturated due to an upper bound on performance, DTP still prioritizes tasks with lesser accuracy (as it has lower performance than others). While DTP has overall higher errors, we observe that it provides the near-to-best accuracy of 79.87% for the lower-performing task of gender recognition. While the overall performance of Adaptive Schedule (Jean et al., 2019) is competitive, the performance plateaus out when a task is getting closer to its STL performance. This limits the performance gain, even if there could be a scope for improvement (e.g., landmark localization).

Compared with other scheduling or dropping-based studies, Maninis et al. (2019) suggested attentive single-task scheduling while Chen et al. (2020) suggested Gradient drop. As seen by the performance of Attentive Tasking (Maninis et al., 2019), scheduling single tasks results in close to optimal outcomes for easier tasks (as they are fast learners). However, a difficult task such as T3 of landmark localization remains incomplete. Finally, a dominant task causing negative transfer can have a high gradient, increasing the chances of its gradient propagation in Gradient Drop (Chen et al., 2020). To counter the effect of the dominant task,

DST identifies such tasks by their properties (depth, data count, incompleteness, and stagnation). Then, DST reduces its dominance by dropping them stochastically. Furthermore, occasional switching "ON" of a task with opposite gradient can also assist in regularization, which DST achieves by its stochasticity. Experimentally, Gradient Drop (Chen et al., 2020) has a net error of 11.07% while DST outperforms with a 10.75% error on the AFLW database.

**Table 4:** Classification accuracy (Gender and Spectacles) and normalized error (Coordinates, Pose, and Denoising) on the AFLW test set. (↓) signifies that a lower value is desired, while (↑) signifies that higher value is desired.

| Training Method | Experiment Name | Classification Accuracy (%) (↑) | | Normalized Error (%) (↓) | | | |
|---|---|---|---|---|---|---|---|
| | | T1: Gender | T2: Spectacles | T3: Landmark | T4: Pose | T5: Denoising | Average Error |
| | Individually Trained (STL) | 78.05 ± 0.13 | 96.82 ± 0.01 | 8.21 ± 0.01 | 0.18 ± 0.01 | 20.83 ± 0.01 | 10.87 ± 0.03 |
| Jointly | MTL | 79.20 ± 0.43 | 96.00 ± 0.22 | 10.31 ± 0.29 | 0.27 ± 0.04 | 21.09 ± 0.01 | 11.29 ± 0.12 |
| Jointly | Asymmetric MTL (Lee et al., 2018) | 78.90 ± 0.59 | 95.95 ± 0.14 | 10.93 ± 0.67 | 0.37 ± 0.05 | 21.17 ± 0.03 | 11.52 ± 0.31 |
| Jointly | GradNorm ($\alpha = 0.5$) (Chen et al., 2018b) | 79.48 ± 0.46 | 95.99 ± 0.26 | 10.73 ± 1.49 | 0.62 ± 0.65 | 21.06 ± 0.14 | 11.39 ± 0.56 |
| Jointly | GradNorm ($\alpha = 1.0$) (Chen et al., 2018b) | **80.16 ± 0.63** | 95.42 ± 0.23 | 13.74 ± 0.03 | 1.87 ± 0.21 | 21.17 ± 0.05 | 12.24 ± 0.11 |
| Jointly | GradNorm ($\alpha = 1.5$) (Chen et al., 2018b) | 79.41 ± 0.45 | 94.05 ± 0.30 | 13.85 ± 0.05 | 13.73 ± 0.04 | 21.28 ± 0.13 | 15.08 ± 0.11 |
| Jointly | LBTW ($\alpha = 0.5$) (Liu et al., 2019a) | 78.87 ± 0.51 | 95.94 ± 0.44 | 8.53 ± 0.17 | 0.30 ± 0.00 | **21.05 ± 0.01** | 11.01 ± 0.02 |
| Jointly | LBTW ($\alpha = 1.0$) (Liu et al., 2019a) | 80.09 ± 0.04 | 95.68 ± 0.04 | 13.99 ± 0.29 | 2.23 ± 0.06 | 21.23 ± 0.00 | 12.34 ± 0.06 |
| Jointly | DTP (Guo et al., 2018) | 79.87 ± 0.24 | 95.48 ± 0.25 | 8.94 ± 1.39 | 0.25 ± 0.02 | 21.20 ± 0.01 | 11.01 ± 0.27 |
| Jointly | Adaptive Schedule (Jean et al., 2019) | 78.77 ± 0.53 | 96.02 ± 0.11 | 8.37 ± 0.04 | **0.24 ± 0.02** | 21.10 ± 0.01 | 10.98 ± 0.09 |
| Jointly | Attentive Tasking (Maninis et al., 2019) | 78.51 ± 0.62 | 96.17 ± 0.29 | 10.22 ± 0.71 | 0.28 ± 0.03 | 21.10 ± 0.02 | 11.38 ± 0.23 |
| Jointly | Gradient Drop (Chen et al., 2020) | 79.27 ± 0.44 | 96.05 ± 0.21 | 9.35 ± 0.17 | 0.26 ± 0.03 | 21.08 ± 0.02 | 11.07 ± 0.16 |
| Jointly | Auto-$\lambda$ (Liu et al., 2022) | 78.16 ± 0.27 | **96.21 ± 0.17** | 7.13 ± 0.41 | 0.26 ± 0.02 | 21.06 ± 0.00 | 10.82 ± 0.11 |
| Jointly | Random ($P_{(k,t)} = 0.50$) | 78.66 ± 0.63 | 95.93 ± 0.23 | 9.87 ± 1.07 | 0.43 ± 0.13 | 21.23 ± 0.13 | 11.39 ± 0.34 |
| Jointly | Random ($P_{(k,t)} = 0.75$) | 77.48 ± 0.59 | 96.14 ± 0.32 | 8.59 ± 0.12 | 0.25 ± 0.01 | 21.09 ± 0.00 | 11.21 ± 0.03 |
| Jointly | DST (Proposed) | 78.26 ± 0.13 | 96.18 ± 0.05 | **6.91 ± 0.32** | **0.24 ± 0.02** | **21.05 ± 0.01** | **10.75 ± 0.08** |

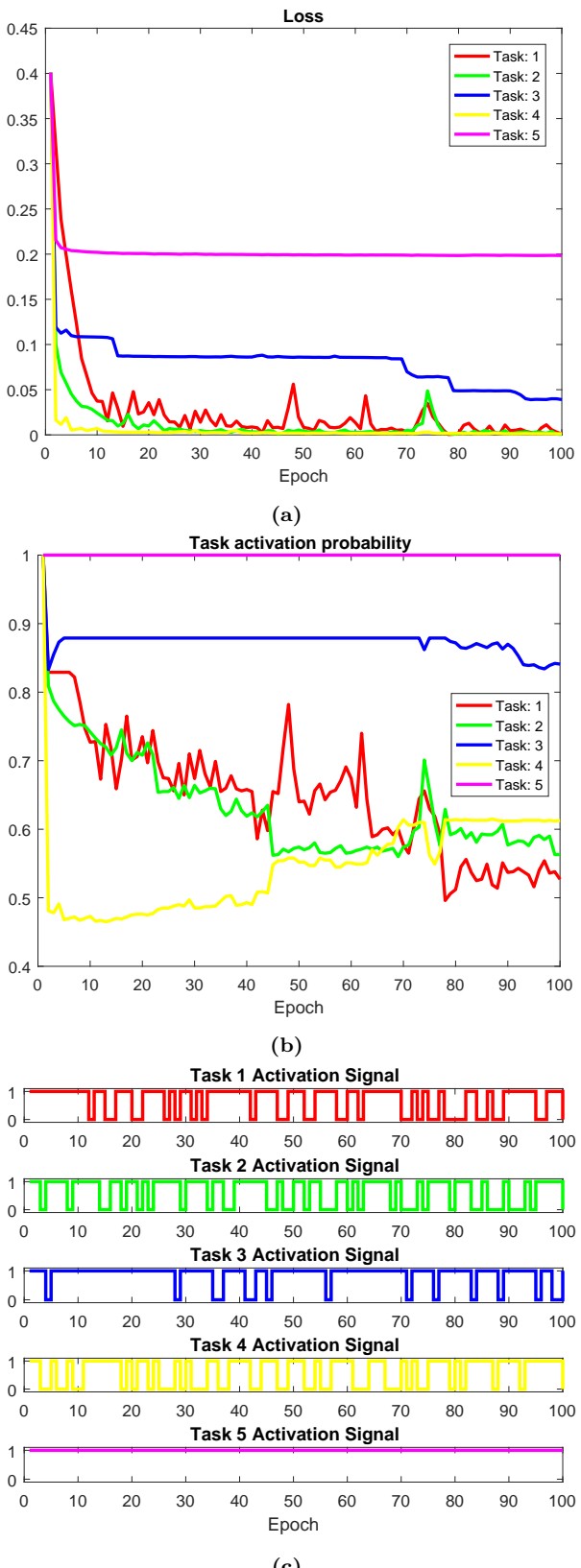

**Figure 6:** For AFLW database, (a) the training loss, (b) the task activation probability, and (c) the task-wise activation using DST algorithm.

**Table 5:** Classification accuracy (Orientation), IoU (Segmentation), and EER (Pairwise matching) on the IIITD MUST test set. (↓) signifies that a lower value is desired, while (↑) signifies that higher value is desired.

| Training Method | Experiment Name | Classification Accuracy (%) (↑) | | IoU (×100) (↑) | | EER (%) (↓) | Average Error (%) (↓) |
| --- | --- | --- | --- | --- | --- | --- | --- |
| | | T1: Orientation Fingerprint | T3: Orientation Latent | T2: Fingerprint Segmentation | T4: Latent Segmentation | T5: Siamese Verification | |
| | Individually Trained (STL) | 82.40 ± 0.58 | 44.24 ± 0.43 | 90.96 ± 1.02 | 79.43 ± 1.17 | 15.83 ± 0.24 | 23.76 ± 0.65 |
| Jointly | MTL | 85.78 ± 2.91 | 55.77 ± 1.24 | 94.37 ± 0.10 | 83.06 ± 0.01 | 22.25 ± 0.33 | 20.66 ± 0.88 |
| Jointly | Asymmetric MTL (Lee et al., 2018) | 86.30 ± 1.17 | 57.81 ± 1.42 | 94.72 ± 0.23 | 83.06 ± 0.77 | 30.47 ± 0.93 | 21.72 ± 1.03 |
| Jointly | GradNorm ($\alpha = 0.5$) (Chen et al., 2018b) | 88.66 ± 0.33 | 58.01 ± 0.90 | 94.62 ± 0.11 | **83.35 ± 0.21** | 21.24 ± 0.80 | 19.32 ± 0.21 |
| Jointly | GradNorm ($\alpha = 1.0$) (Chen et al., 2018b) | 76.47 ± 5.62 | 50.28 ± 1.10 | 94.71 ± 0.32 | 82.94 ± 0.62 | 35.43 ± 0.45 | 26.21 ± 1.07 |
| Jointly | GradNorm ($\alpha = 1.5$) (Chen et al., 2018b) | 45.88 ± 4.90 | 24.45 ± 2.76 | 93.14 ± 0.49 | 80.97 ± 0.35 | 33.98 ± 1.04 | 37.91 ± 3.30 |
| Jointly | LBTW ($\alpha = 0.5$) (Liu et al., 2019a) | 85.90 ± 0.66 | 55.92 ± 0.90 | 94.35 ± 0.28 | 83.12 ± 0.49 | 21.65 ± 1.55 | 20.47 ± 0.20 |
| Jointly | LBTW ($\alpha = 1.0$) (Liu et al., 2019a) | 81.05 ± 2.93 | 51.95 ± 0.47 | 94.39 ± 0.41 | 80.67 ± 3.02 | 37.68 ± 0.30 | 25.92 ± 0.19 |
| Jointly | DTP (Guo et al., 2018) | 85.12 ± 0.34 | 57.67 ± 0.64 | 93.61 ± 0.16 | 81.07 ± 0.06 | 22.36 ± 0.10 | 20.98 ± 0.20 |
| Jointly | Adaptive Schedule (Jean et al., 2019) | 86.42 ± 2.15 | 56.61 ± 1.14 | 94.44 ± 0.15 | 82.70 ± 0.57 | 23.30 ± 1.50 | 20.63 ± 1.10 |
| Jointly | Attentive Tasking (Maninis et al., 2019) | 88.93 ± 0.78 | 59.40 ± 1.19 | 94.37 ± 0.01 | 82.48 ± 0.51 | 21.69 ± 0.82 | 19.30 ± 0.38 |
| Jointly | Gradient Drop (Chen et al., 2020) | 86.57 ± 3.07 | 57.18 ± 0.81 | 94.29 ± 0.23 | 83.12 ± 0.48 | 21.25 ± 1.22 | 20.02 ± 0.92 |
| Jointly | Auto-$\lambda$ (Liu et al., 2022) | 88.35 ± 0.63 | 59.31 ± 0.94 | 94.01 ± 0.08 | 81.69 ± 0.64 | 20.72 ± 0.13 | 19.47 ± 0.12 |
| Jointly | Random ($P_{(k,t)} = 0.50$) | 88.43 ± 4.27 | 58.23 ± 0.32 | 93.50 ± 0.48 | 79.80 ± 1.17 | 23.37 ± 0.14 | 20.68 ± 0.62 |
| Jointly | Random ($P_{(k,t)} = 0.75$) | 86.86 ± 0.76 | 59.54 ± 0.02 | 94.39 ± 0.05 | 82.74 ± 1.05 | 22.22 ± 0.39 | 19.74 ± 0.45 |
| Jointly | DST (Proposed) | **89.14 ± 1.29** | **59.66 ± 0.75** | 94.45 ± 0.32 | 82.69 ± 1.18 | **20.28 ± 0.53** | **18.87 ± 0.42** |

**Table 6:** Segmentation performance (IoU) for DST compared against different encoder-decoder deep architectures.

| | SegNet (Badrinarayanan et al., 2017) | UNet (Ronneberger et al., 2015) | MTL | DST |
|---|---|---|---|---|
| Fingerprint | 0.91±0.01 | 0.92±0.01 | **0.94±0.00** | **0.94±0.00** |
| Latent | 0.79±0.01 | 0.80±0.02 | **0.83±0.00** | **0.83±0.01** |

## 4.2 Bilateral MTL: Fingerprints

When MTL is performed for different tasks with input as pair of fingerprint images, the results are shown in Table 5, followed by segmentation task results in Table 6. Visually, prediction outputs for latent and fingerprint segmentation is shown in Figure 12 and 13 of the appendix and briefly in Figure 7. Figure 14 in appendix shows the ROC curve for the fifth task of latent fingerprint comparison.

Under bilateral MTL with application on fingerprints, the fifth task of pairwise matching has the largest number of annotated ground-truth samples. Training other tasks in presence of pairwise matching task improves their performances. However, the fifth task (pairwise matching) undergoes a negative transfer due to the segmentation tasks, which have far lower samples. The DST algorithm ensures performance gain in orientation and verification tasks with a negligible drop in the performance of latent fingerprint segmentation task (T4). Overall, with improvement in four tasks, the proposed DST algorithm reports the least overall error across algorithms. Visually, as seen in Figure 7, DST gives good segmentation results for fingerprint segmentation (T2) and sufficient latent fingerprint segmentation results (T4). We compare results of the DST algorithm with MTL and other encoder-decoder algorithms for segmentation, as shown in Table 6. We observe that the proposed DST algorithm reports the best IoU for fingerprint segmentation. Further, similar to MTL, the DST algorithm also overpowers the standard SegNet and U-Net for latent fingerprint segmentation. Hence, with a minimal reduction in the performance of latent segmentation, the DST algorithm limits negative transfer. The training graphs of DST for fingerprints are shown in Figure 11 in the appendix.

The proposed DST algorithm has the overall best performance of 18.87% compared to other task prioritization algorithms for fingerprint applications. Similar to faces, GradNorm and LBTW provide competitive performances when $\alpha = 0.5$. However, they report higher errors for $\alpha \geq 1$, where possibly the solution drifted away from the optima. Asymmetric MTL, while considering easier tasks as reliable, does give one of the best results for segmentation tasks. However, it has the highest overall error due to sub-par performance on the difficult task of verification (T5). DTP aims to prioritize lower-performing tasks. Thus, DTP provides a near-to-best performance of 57.67% for latent fingerprint orientation classification (T3) as it is the worst-performing task across five tasks of fingerprints (when trained in MTL setting). However, due to T3, well-performing tasks T1, T2, and T4 observe a higher drop in performance. Hence, DTP reports an overall error of 20.98% compared to 18.87% by DST. Attentive single-tasking and Auto-$\lambda$ provide one of the best performances after the proposed DST algorithm. This can be attributed to better disentangling and reduced interference, as unlike faces and character recognition, the tasks are more diverse in fingerprints. Gradient Drop provides an overall error of 20.02%. While T2, T4, and T5 are comparable, DST outperforms due to higher accuracy in T1 and T3. This possibly results from more "ON" time for T3 (latent orientation) in DST, which is similar to T1 (fingerprint orientation) and improves its performance as well(refer to Figure 11(c) of the appendix).

## 4.3 Unilateral MTL: Character Recognition

We illustrate the results of multi-task character recognition using the Omniglot dataset, which is commonly used to highlight performance on a high number of tasks in MTL. In this study, since face and fingerprint had architecture similar to VGG16, we also utilized VGG (encoder only) based architecture for this application. Given all the tasks are classification tasks for Omniglot character recognition, all tasks are placed at the same depth. A VGG encoder is used as the shared backbone, followed by a branch for each task after the last encoder layer to classify a character in the MTL setup. The branch consists of three dense layers, with the

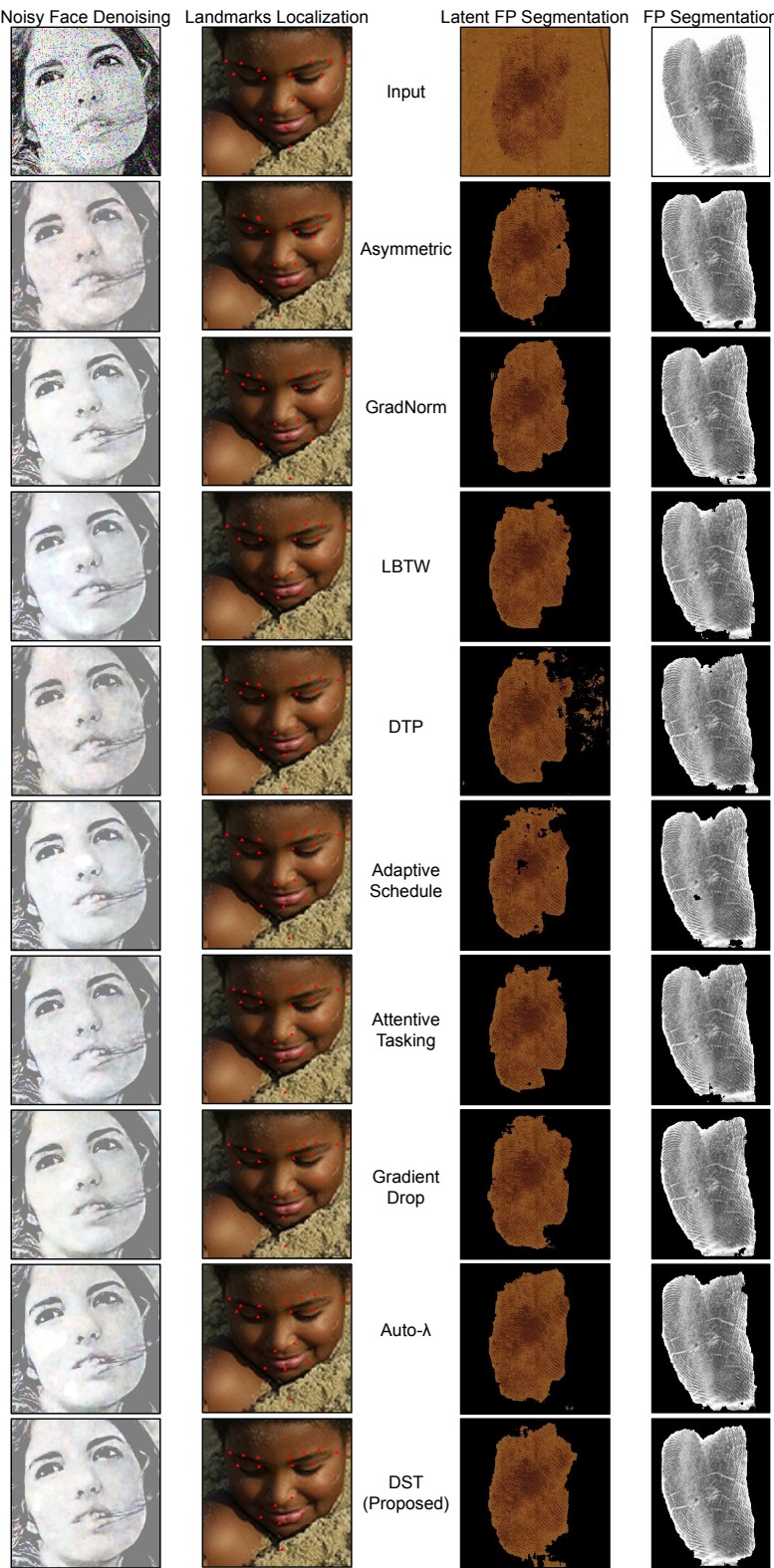

**Figure 7:** Sample outputs from the proposed DST and other comparative algorithms for face denoising, landmark localization, and segmentation tasks. Additional instances are shown in the appendix.

**Table 7:** Average error (%)(↓) on the Omniglot dataset.

| Algorithm | Avg. Error (%) |
|---|---|
| MTL | 11.26 |
| Asymmetric MTL (Lee et al., 2018) | 10.81 |
| GradNorm ($\alpha = 0.5$) (Chen et al., 2018b) | 10.84 |
| LBTW ($\alpha = 0.5$) (Liu et al., 2019a) | 10.59 |
| DTP (Guo et al., 2018) | 10.77 |
| Adaptive Schedule (Jean et al., 2019) | 10.74 |
| Attentive Tasking (Maninis et al., 2019) | 10.67 |
| Gradient Drop (Chen et al., 2020) | 10.61 |
| Auto-$\lambda$ (Liu et al., 2022) | 10.52 |
| **DST (Proposed)** | 10.41 |

**Table 8:** Ablation results on AFLW dataset. T1: gender, T2: spectacles, T3: landmark, T4: pose, & T5: denoising.

| Details | Accuracy (%)(↑) | | Normalized Error (%)(↓) | | | |
|---|---|---|---|---|---|---|
| | **T1** | **T2** | **T3** | **T4** | **T5** | **Avg.** |
| **Proposed** | 78.26 | 96.18 | 6.91 | 0.24 | 21.05 | 10.75 |
| Small MTL | 70.31 | 96.11 | 5.69 | 0.29 | 25.56 | 13.02 |
| Small DST | 70.51 | 96.47 | 5.28 | 0.25 | 24.61 | 12.63 |
| Early stop | 77.97 | 91.71 | 8.99 | 0.26 | 21.05 | 12.12 |
| Just $\mathcal{P}_{(u,k,t)}$ | 79.43 | 96.23 | 8.97 | 0.47 | 21.02 | 10.96 |
| $\mathcal{P}'_{(c,t)}$ | 77.89 | 95.67 | 8.47 | 0.29 | 21.09 | 11.26 |

dense output layer having units equal to the number of classes. As mentioned earlier, the shared encoder in MTL is initialized with VGG16 (Simonyan & Zisserman, 2014) pre-trained weights of the ImageNet dataset.

Using this architecture, MTL yields an error of 11.26%. Table 7 also shows the performance of other task prioritization algorithms. Compared to just 10.41% error by the proposed DST algorithm, other task prioritization algorithms have a relatively higher error in the range of 10.52% to 10.84%. For reference, with a fixed ResNet18 architecture, Prellberg & Kramer (2020) illustrated the average baseline error on Omniglot as ~10.70%.

### 4.4 Ablation study

The proposed DST algorithm involves multiple variability points that need to be tested to validate the effectiveness of DST. Table 8 and Table 9 highlight ablation results under different settings. With the ablation study, we check the impact of weight initialization, network depths, data count, $w_t$ in Eq. 17, varying $\lambda$ (from Eq. 8) and, consequently, the effect of each of the four metrics. We also check a closely possible extension of DST, where the tasks get dropped with a fixed static probability. Each of these ablations is presented in this subsection. These ablation experiments are conducted on the AFLW dataset.

#### 4.4.1 Random Drop

A static random drop is closer to the dynamic task dropping of DST for comparison. To perform ablation and check if a random drop can achieve a good performance, we drop tasks with a probability of 0.25 and 0.50. This implies that with a throughout fixed activation probability $P_{(k,t)}$, the tasks are stochastically dropped with a constant probability $1 - P_{(k,t)}$. Static dropping gives slightly better performance than MTL, but not that of DST, as highlighted in the bottom three rows of Table 4 and 5. Random drop reports the second-best results after the DST algorithm for a few tasks. For instance, T2 of AFLW (wearing spectacles?) has just 0.04% lower performance for random drop than DST. Likewise, T4 of latent fingerprint segmentation also has a difference of 0.05% in IoU. One reason could be the fact that for these tasks, the DST algorithm has $E(P_{(k,t)})$ closer to the static random drop probability. However, of overall performance, DST outperforms static random drop.

**Table 9:** Ablation study on the AFLW (Koestinger et al., 2011) test set by leaving out or keeping certain metrics. Classification accuracy (Gender and Spectacles) and normalized error (Coordinates, Pose, and Denoising) reported different probability weights. A (↓) signifies that a lower value is desired, while (↑) signifies that higher value is desired. Only $\mu$ values reported here in interest of space.

| Details | Regularization | Individual Probability Weights | | | | | Classification Accuracy (%) (↑) | | Normalized Error (%) (↓) | | | |
|---|---|---|---|---|---|---|---|---|---|---|---|---|
| | | $\lambda_d$ | $\lambda_c$ | $\lambda_u$ | $\lambda_r$ | $\lambda_b$ | T1: Gender | T2: Spectacles | T3: Coordinates | T4: Pose | T5: Denoising | Average Error |
| Traditional MTL | | | | | | | 79.20 | 96.00 | 10.31 | 0.27 | 21.09 | 11.29 |
| Leave-one metric out | ✓ | **0.00** | 0.25 | 0.25 | 0.25 | 0.25 | 78.37 | 96.37 | 9.11 | 0.26 | 21.10 | 11.15 |
| | ✓ | 0.25 | **0.00** | 0.25 | 0.25 | 0.25 | 78.86 | 96.18 | 8.86 | 0.29 | 21.07 | 11.04 |
| | ✓ | 0.25 | 0.25 | **0.00** | 0.25 | 0.25 | 78.85 | 96.10 | 9.89 | 0.27 | 21.07 | 11.25 |
| | ✓ | 0.25 | 0.25 | 0.25 | **0.00** | 0.25 | 79.14 | 96.05 | 8.99 | 0.29 | 21.06 | 11.03 |
| | ✗ | 0.25 | 0.25 | 0.25 | 0.25 | **0.00** | 79.10 | 96.38 | 9.80 | 0.29 | 21.05 | 11.13 |
| Only static metrics | ✗ | **1.00** | 0.00 | 0.00 | 0.00 | 0.00 | 79.06 | 96.22 | 9.65 | 0.34 | 21.09 | 11.16 |
| | ✗ | 0.00 | **1.00** | 0.00 | 0.00 | 0.00 | 77.92 | 96.45 | 9.60 | 0.32 | 21.10 | 11.33 |
| Only dynamic metrics | ✗ | 0.00 | 0.00 | **1.00** | 0.00 | 0.00 | 79.43 | 96.23 | 8.97 | 0.47 | 21.02 | 10.96 |
| | ✗ | 0.00 | 0.00 | 0.00 | **1.00** | 0.00 | 79.04 | 96.27 | 8.75 | 0.27 | 21.06 | 10.96 |
| Varying weights w/o regularization | ✗ | 0.20 | 0.20 | 0.30 | 0.30 | 0.00 | 78.14 | 96.30 | 8.41 | 0.30 | 21.09 | 11.07 |
| | ✗ | 0.10 | 0.10 | 0.40 | 0.40 | 0.00 | 79.42 | 95.93 | 8.68 | 0.27 | 21.05 | 10.93 |
| | ✗ | 0.40 | 0.40 | 0.10 | 0.10 | 0.00 | 78.56 | 96.14 | 10.90 | 0.34 | 21.07 | 11.52 |
| Varying weights with regularization | ✓ | 0.15 | 0.15 | 0.30 | 0.30 | 0.10 | 78.36 | 96.25 | 8.39 | 0.31 | 21.06 | 11.03 |
| | ✓ | 0.30 | 0.30 | 0.15 | 0.15 | 0.10 | 78.96 | 96.11 | 9.82 | 0.29 | 21.07 | 11.22 |
| Proposed | ✓ | 0.20 | 0.20 | 0.20 | 0.20 | 0.20 | 78.26 | 96.18 | 6.91 | 0.24 | 21.05 | 10.75 |

### 4.4.2 Network Initialization

We used encoder with VGG16 ImageNet pre-trained weights for stable initialization (to mitigate high initial loss ($V_{(1,t)}$). Alternatively, we also estimate initial loss with $N = 10$ different initializations and observe that it slightly reduces the effect of randomness. It yields similar average error of 10.70%.

### 4.4.3 Varying Network Depth

Negative transfer is more likely when a task creates dominance due to being shallower. To show stability for different network structure, T1 FC branch is preponed to $7^{th}$ conv instead of original $13^{th}$. Position of other tasks remain the same. With MTL (Row-2, Table 8), we observe an adverse effect on the deepest task T5. T1 gets deteriorated due to information loss (larger GAP resolution). T3 got improved with less interference in learning at encoder. With proposed DST (Row-3, Table 8), the dominance of shallower task T1 gets reduced on the deepest task T5. Overall, with other network architectures, we observe improved performance with DST.

### 4.4.4 Overfitting

A cause for negative transfer could be overfitting. To counter overfitting, we: (i) early stop, or (ii) probabilistically drop early completed tasks (only $\mathcal{P}_{(u,k,t)}$ as metric). For first case, completed tasks are early stopped when its MTL loss value reaches its optimal STL loss value (Row 4, Table 8). Here, classification tasks suffer as other regression tasks (T3, T5) stay longer to make undesirable weight updates for T1 and T2. Secondly, with only $\mathcal{P}_{(u,k,t)}$, we observe overall sub-par performance with T3 being the most affected (Row 5, Table 8). T3 got limited $ON$ time despite its stagnation (due to absence of $\mathcal{P}_{(r,k,t)}$).

### 4.4.5 Inverse case of data count

We also perform ablation for the inverse case where a task with lower ground-truth labeled samples is preferred over a task with more labeled samples (against the original assumption of Eq. 2 where a task with more labeled samples is preferred). If the performance turns out to be better than the proposed DST, this could mean that a task with lower ground-truth labeled samples leads to positive transfer. The selection criterion is updated as follows:

$$\mathcal{P'}_{(c,t)} = \frac{\min_{1 \leq t \leq K}(c_t)}{c_t}. \tag{22}$$

The results from the above formula are reported in row 6 of Table 8. We observed a higher average error of 11.26% than 10.75% from the proposed DST. Hence, our original assumption holds that tasks with fewer samples can hamper generalizability while higher sample tasks improve generalization (also shown by (Wu et al., 2020)).

### 4.4.6 $w_t$ in Eq. 17

$w_t$ scales losses to a standard initial value such that task-wise losses are in the same range for Figure 6a. Removing it has a minimal effect on the DST algorithm as both task incompleteness (Section 2.1.3) and stagnation (Section 2.1.4) operate on loss ratios. With $w_t$, the error of MTL reduces by 0.54% while without $w_t$, the reduction is by 0.40% only.

### 4.4.7 Varying $\lambda$ and Metrics

A detailed ablation on varying $\lambda$ values in Eq. 8 is shown in Table 9. It also includes keeping certain $\lambda = 0$ (not considering a metric) and keeping a specific $\lambda = 1$ (considering only one metric). As we see in the rows titled "*Only dynamic metrics*qq, the greatest contribution towards performance is due to dynamic parameters ($\mathcal{P}_{(u,k,t)}$ and $\mathcal{P}_{(r,k,t)}$). In these cases, one of the lowest errors of 10.96% is observed. While other static metrics encode intrinsic properties of model and amount of training labels, $\mathcal{P}_{(u,k,t)}$ and $\mathcal{P}_{(r,k,t)}$ dynamically favors tasks during training that require more computational cycles. We observe the same behavior when $\lambda_u$ and $\lambda_r$ are relatively given higher weights, and an even lower error of 10.93% is seen.

## 5 Conclusion

Due to joint optimization in Multitask Learning, some task(s) may face a *negative transfer*. We propose *Dropped Scheduled Task* (DST) algorithm to reduce the impact of negative transfer. Based on the scheduling probability, the DST algorithm gives computation cycles to specific tasks while the others are "*dropped*". For each task, the scheduling probability is decided based on four different metrics. These metrics rely on: (i) task depth, (ii) the number of ground-truth samples per task, (iii) task training completion, and (iv) task stagnancy. For experimental results, three MTL applications are chosen related to faces, (latent) fingerprints, and character recognition. Due to the occasional task *dropping* of learned or dominating tasks, more computation cycles are given to stagnant/slower/weaker tasks, helping them to move towards their minima. In the process, the *dropped* task(s) refrain from overfitting, resulting in a generalizable solution. Overall, the results show minimum negative transfer and overall least error across different algorithms and tasks.

This research focuses on the metrics defined in the DST algorithm for vanilla MTL. As for future implications of this study, these metrics can also have broader implications and use cases. For instance, task incompleteness and task stagnation metrics can be extended for the case of single-task classification models, where the aim is to increase the performance for a weaker class. Similarly, the metric on task depth and training sample count can become vital in MTL with auxiliary tasks. In such a scenario, these metrics can be used as it is or altered to prioritize the main task more compared to other auxiliary tasks.

## Acknowledgments

A. Malhotra acknowledges IIT Jodhpur for hosting him during this research. A. Malhotra is partially supported through Visvesvaraya Ph.D. Scheme by MEITY, Govt. of India. M. Vatsa is supported through the Swarnajayanti Fellowship by the Government of India.

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

# A   Appendix

---

**Algorithm 1:** Sampling Task-wise activation bits (ON/OFF) with Dropped Scheduled Task (DST) Algorithm

---

**Result:** $G_{(k,:)}$: a 1×K vector of independent Bernoulli random variables for activating/dropping individual tasks

Given K tasks, task-wise network depth $d_t$, and task-wise annotated sample count $c_t$

Initialize network weights $W$

Initialize a value for $\beta \in [0, 1]$

Initialize $\lambda_d$, $\lambda_c$, $\lambda_u$, $\lambda_r$, $\lambda_b \in [0, 1]$ such that $\lambda_d + \lambda_c + \lambda_u + \lambda_r + \lambda_b = 1$

**for** *each task t* **do**

    Set the probability based on network depth $P_{(d,t)} = \frac{d_t}{\max\limits_{1 \leq t \leq K} (d_t)}$

    Set the probability based on training sample count $P_{(c,t)} = \frac{c_t}{\max\limits_{1 \leq t \leq K} (c_t)}$

    Set regularization probability $P(b,t) = 1$

**end**

**for** *each epoch k* **do**

    **for** *each task t* **do**

        **if** $k == 1$ **then**

            Calculate task-wise loss value without weight update and store as $V_{(1,t)}$

            Set $P_{(u,k,t)} = 1$

            Set $P_{(r,k,t)} = 1$

        **else**

            $I_{(k,t)} = \frac{V_{(k,t)}}{V_{(1,t)}}$

            Calculate task incompletness probability $P_{(u,k,t)} = \min(1, \frac{I_{(k,t)}}{E(I_{(k)})})$

            Calculate local rate of change $R_{(k,t)} = \frac{1}{P_{(u,k,t)}} \times \frac{V_{(k,t)} - V_{(k-1,t)}}{V_{(k-1,t)}}$

            **if** $k == 2$ **then**

                Set global rate of change $R'_{(k,t)} = R_{(k,t)}$

            **else**

                Update global rate of change $R'_{(k,t)} = \beta R_{(k,t)} + (1 - \beta)R'_{(k-1,t)}$

            **end**

            Calculate task stagnancy probability $P_{(r,k,t)} = \min(1, \frac{E(R'_{(k)})}{R'_{(k,t)}})$

        **end**

        Calculate the final probability for task scheduling

        $P_{(k,t)} = \lambda_d P_{(d,t)} + \lambda_c P_{(c,t)} + \lambda_u P_{(u,k,t)} + \lambda_r P_{(r,k,t)} + \lambda_b P_{(b,t)}$

        Sample task ON/OFF bit $G_{(k,t)} = \text{Bernoulli}(P_{(k,t)})$

    **end**

**end**

---

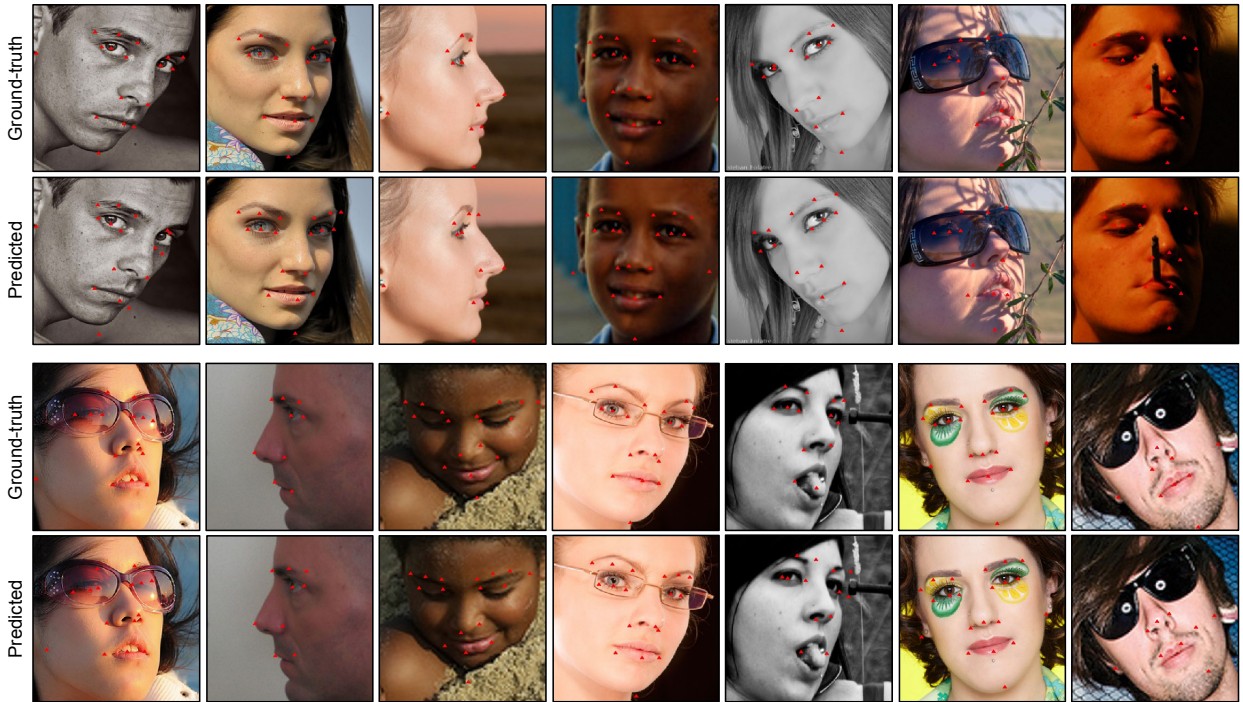

**Figure 8:** Landmark localization results from the DST algorithm for the AFLW database.

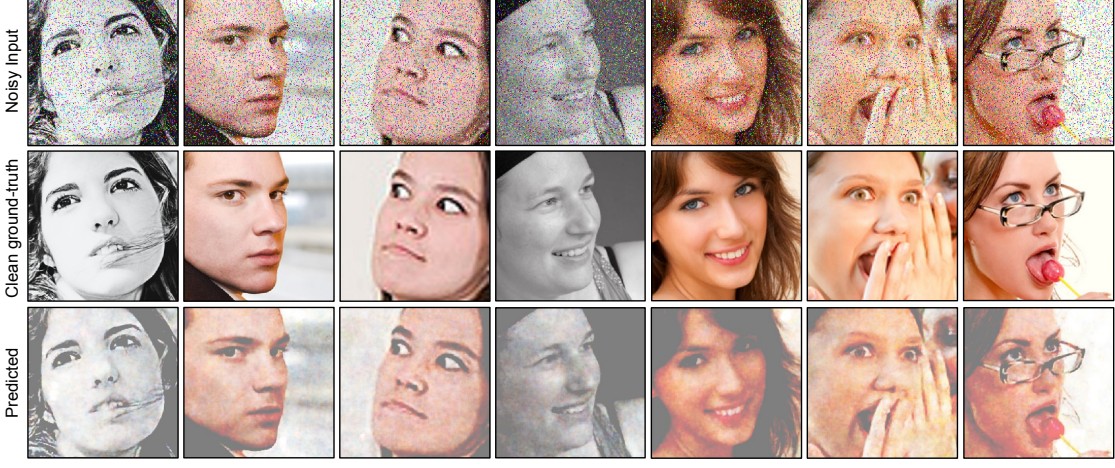

**Figure 9:** Image denoising results from the DST algorithm for the AFLW database.

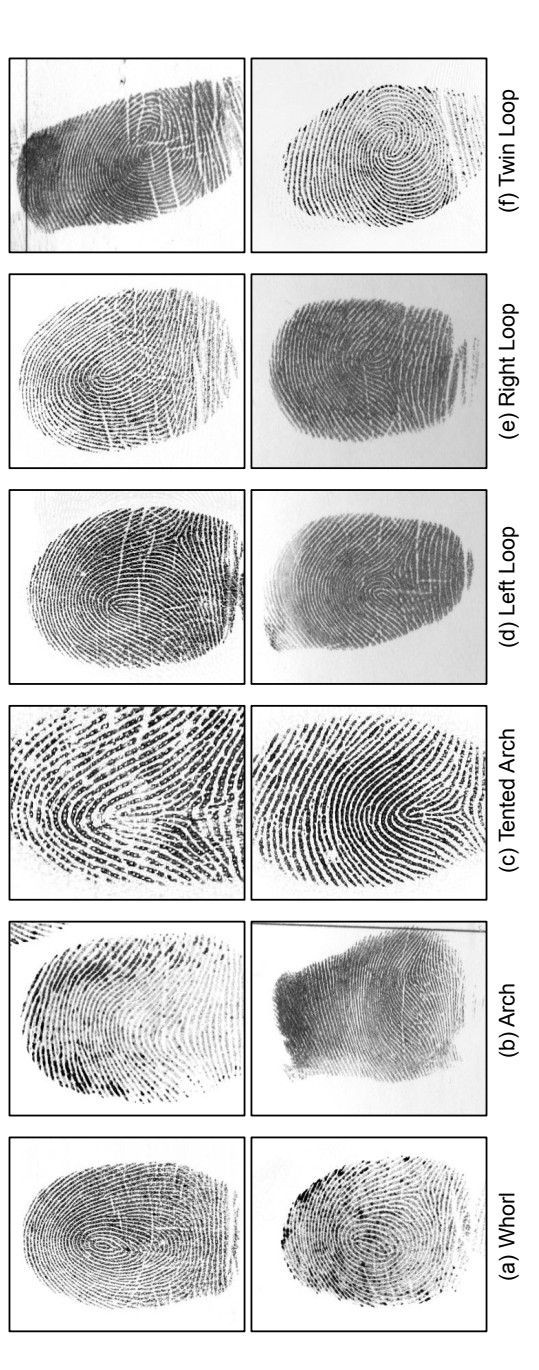

**Figure 10:** The six types of fingerprint orientation patterns (also known as level-1 features) used for six class classification tasks.

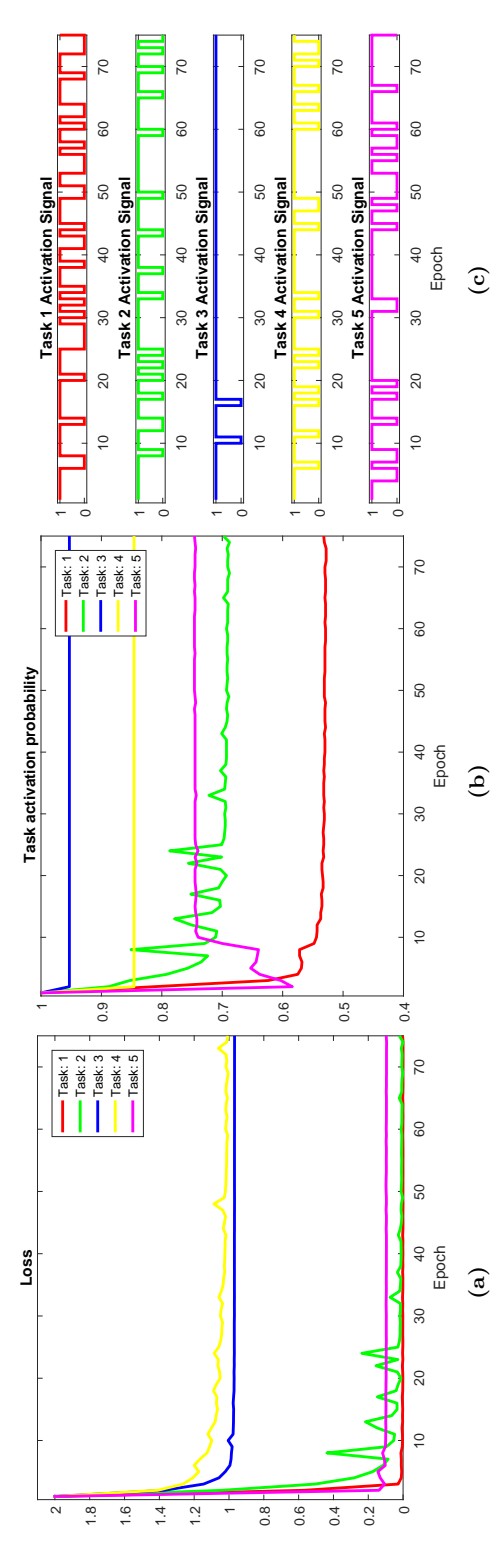

**Figure 11:** For the consolidated latent fingerprint database, the task-wise (a) training loss, (b) activation probability, and (c) activation using DST algorithm.

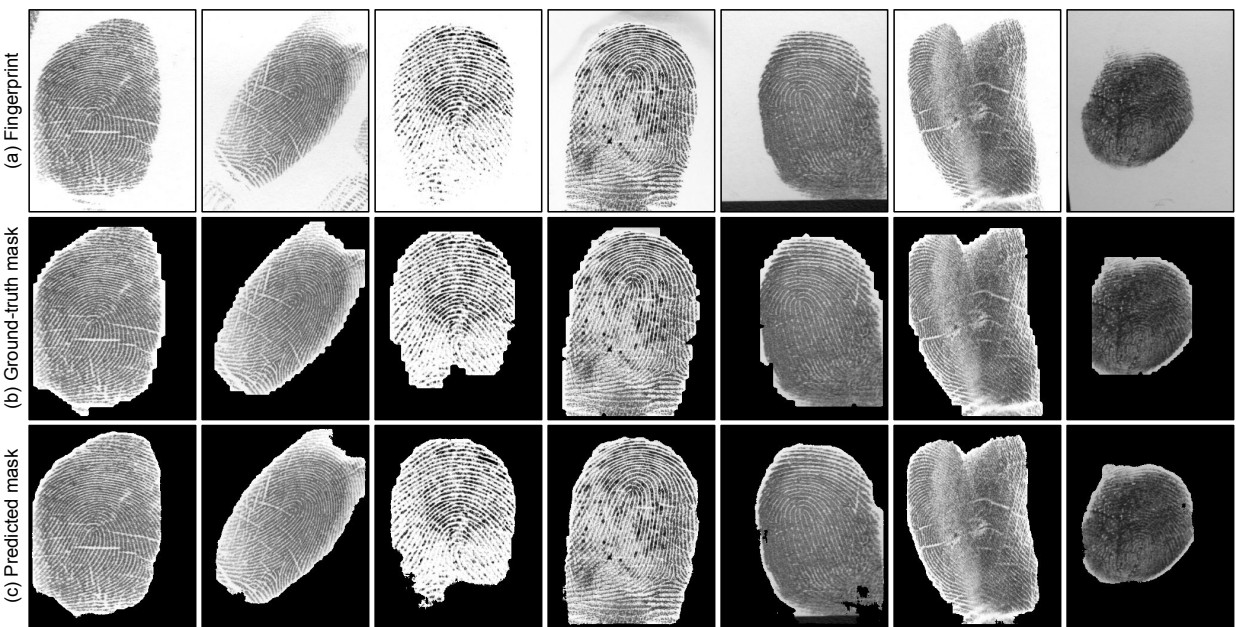

**Figure 12:** Few sample outputs for fingerprint segmentation (task 2), compared against the ground-truth masks.

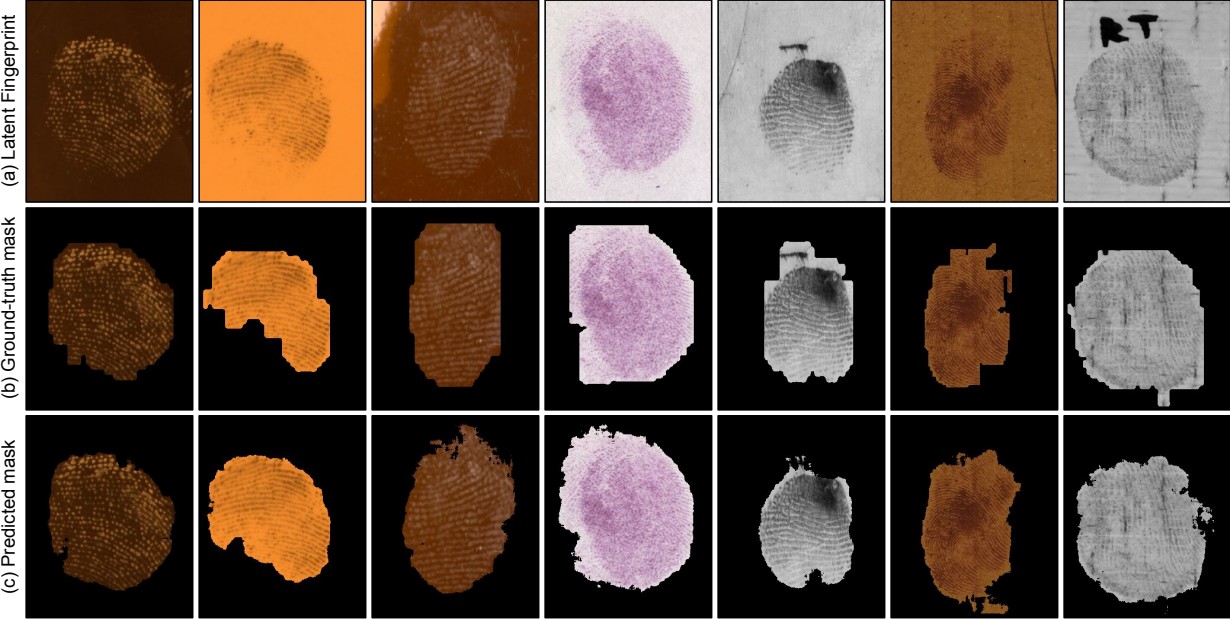

**Figure 13:** Few sample outputs for latent fingerprint segmentation (task 4), compared against the ground-truth masks.

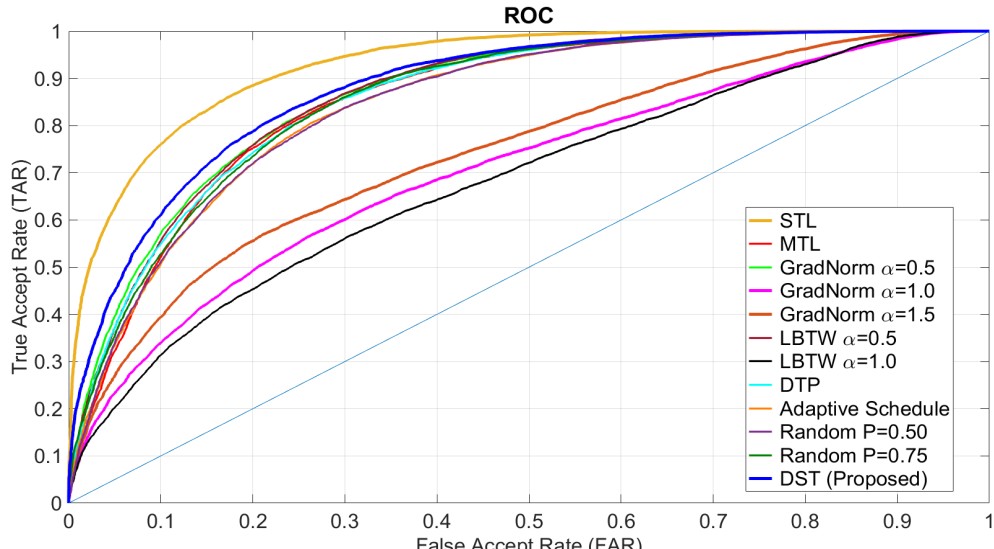

**Figure 14:** ROC curve for latent fingerprint matching (task 5).

