# OpenReview forum: "Dropped Scheduled Task: Mitigating Negative Transfer in Multi-task Learning using Dynamic Task Dropping"
_TMLR — Accepted by TMLR_

### Review · Reviewer_P6J7 · 2022-10-05

**Summary Of Contributions:**

The authors propose a number of heuristic rules for dynamically removing the influence of specific tasks during multi-task learning (MTL). This is claimed to limit the effect of "negative transfer"---situations where the learning of one task contributes adversely to the learning of another. The heuristics are used to define a probability that the influence of a task is paused during one epoch of training. In a series of MTL experiments, the authors find that the proposed method compares favourably to baselines from the literature.

**Requested Changes:**

The concept of "negative transfer" is fundamental to the paper yet never formally defined. The authors cite Wang et al (2019) who attempt to define negative transfer, but, as a reader less familiar with the details of the MTL literature, I am not sure whether this definition is agreed upon in the community. Nevertheless, I think it would be appropriate to state _a_ definition in the paper.

In Figure 1, negative transfer is illustrated in terms of transfer between pairs of tasks A and B in the context of another task C, where A can have a negative effect on B and B has a positive effect on C. This type of transfer is never defined in the context of multiple (>2) tasks, and never estimated from what I can tell. A clarification would be useful.

In Figure 7, examples of the output from the DST algorithm are given, but no examples from other algorithms. This makes comparison difficult. Perhaps a few examples from other algorithms could be added?

**Strengths And Weaknesses:**

Strengths:
The method is simple to implement and yields good results on considered tasks.

Weakness:
The approach targets only limitations of MTL that are due to the training procedure, which could be overcome by scheduling the tasks. More precisely, it does not reason about potential incompatibilities between tasks which would render joint training a lost cause. In other words, should we expect there to exist _any_ algorithm without negative transfer? This limitation is reflected in the four metrics used for scheduling, none of which are directly concerned with the compatibility of learning targets. Naturally, such compatibility is hard to measure a priori, but in its absence it is difficult to assess _when_ we should expect the proposed method to work, and why.

The justification for the proposed approach is very informal, for example, by referring broadly to concepts like vanishing gradients due to network depth. Similarly, "task incompleteness" is defined as the ratio of the loss in the first epoch and the loss in the kth epoch, normalized by on the "expectation" of this ratio over tasks. It is is not clear to me that the different losses in 2.2 have comparable scales/units which would make such a normalization justified.

(+ a few clarity issues in the next section)

Questions:
Related to the previous point, the ablation study on AFLW indicates that using only p(u,k,t) (task incompleteness) for scheduling achieves performance close to the full model. A potential take-away is the most important factor is to make sure that no task is left behind. Does this hold also for other datasets/MTL tasks? Do other MTL algorithms target this idea? In some sense, it appears opposite to the curriculum learning idea of focusing on easier tasks in the beginning of training.

---

> ### Author Response · Authors · 2022-11-17
> **Detailed response to comments of Reviewer P6J7 (Part 1: Addressing weaknesses)**
>
> 1. Comment 1 (from weakness): The authors thank the reviewer for pointing out the task’s compatibility in the multi-task learning setting. Determining tasks that positively influence the performance of other tasks should be trained together. As mentioned by the reviewer, evaluating this compatibility (or task-relatedness) is hard to measure a priori and can require extensive domain knowledge. However, quantifying task-relatedness is beyond the scope of this research. This research aims to make training efficient under the multi-task learning setup given the tasks, positions, and training samples. Furthermore, the tasks are typically placed at different places (depths) in the network [1] or utilize task hierarchies [2] and task grouping [3] to address compatibility. That’s where the metric on task depth is valuable, which can control the effects of gradients of different scales at different depths. Furthermore, with varying combinations of tasks selected for training at each epoch, the occasional dropping of non-compatible tasks can implicitly enhance the overall performance of the tasks. Additionally, in the subsequent responses, we have defined negative transfer and the details associated with it.
>
> [1] Sanh et al. A hierarchical multi-task approach for learning embeddings from semantic tasks. AAAI, 2019.
>
> [2] Xue et al. Multi-Task Learning for Classification with Dirichlet Process Priors. JMLR 2007.
>
> [3] Fifty et al. Efficiently identifying task groupings for multi-task learning. Neurips 2021.
>
>
> 2. Comment 2 (from weakness):
>
> (2A.) To formalize the problem statement in the paper, we have rewritten various parts of the paper. To begin with, we have added a new Section 1.1 on Page 2, titled “Background and Problem Formulation”. Here, we formally define MTL and the problem of negative transfer. Next, we have also rewritten Sections 2.1.1 and 2.1.4 for better understanding.
>
> (2B.) The scale of losses in task incompleteness: To ensure that the drop probability based on task incompleteness, P(u,k,t), in Eq. 4 is consistent, we first normalize the input of Eq. 4 in Eq. 3 to handle losses’ scale variation. In Eq. 3, the incompleteness ratio I(k,t) is a ratio calculated for each task t compared to its initial loss. This yields a value between 0 to 1 for each task. While the input to Eq. 3 might have different scales across different tasks t, the normalization by each task’s respective loss at the first epoch V(1,t) results in  I(k,t) 𝝐 [0,1]. Hence, two tasks T1 and T2 can have a different initial beginning loss value of V(1,1)=5.0 and V(1,2) = 2.0, respectively. However, say after the 10th epoch, assume the values reduce to V(10,1)=4.0 and V(10,2)=1.6 for T1 and T2, respectively. Eq. 3 then calculates the incompleteness ratio I(10,t) as 0.8 for both tasks, which is fed as input to Eq. 4. Finally, both the tasks get equal drop probability in Eq. 4, that is, P(u,10,1) == P(u,10,2).
>
> (2C.) We have rewritten the complete section 2.1.4 on Page 7 of the updated paper for better clarity.
>
> 3. Comment 3 (from weakness): P(u,k,t) plays an essential role in the proposed MTL optimization method alongside the other three proposed metrics. Individually, it encodes task incompleteness, due to which it prioritizes the left-out tasks. Within P(r,k,t), it helps prioritize incomplete tasks which are stagnant. While it is correct to say that P(u,k,t) ensures that no task is left behind, it does not counter the curriculum learning idea where the focus is on easier tasks at the beginning of training. In the proposed DST MTL algorithm, the easier task dominates and gets trained themselves in the initial training phase. However, difficult tasks are left behind and require more training epochs in the later training phase. Hence, difficult tasks are given more training cycles using DST in the absence of easier tasks (where easier tasks are dropped). The same is illustrated in Figure 6(a), where the easier tasks (Tasks 1, 2, and 4) get trained within the first ten epochs. Post the tenth epoch, the DST algorithm de-prioritizes easier tasks. Hence, the activation probability of easier tasks (Tasks 1, 2, and 4) starts reducing, as seen in Figure 6(b).
>
> The proposed DST algorithm, used to optimize MTL training, remains coherent with other MTL algorithms presented in the literature. Most MTL algorithms that aim to improve training procedures aim to prioritize the left-out tasks, assuming that easier tasks do not require any special assistance. For instance, GradNorm and LBTW prioritized left-out tasks whose losses have not reduced significantly with their respective starting loss. Auto-Lambda (Liu et al.) also utilized validation loss to compute task weights. DTP and Adaptive Schedule also prioritized left-out tasks with the criterion chosen as performance instead of the loss value to compute completeness. To summarize, other MTL algorithms also target the idea of prioritizing left-out tasks for good performance.

---

> ### Author Response · Authors · 2022-11-17
> **Detailed response to comments of Reviewer P6J7 (Part 2: Addressing requested changes)**
>
> 1. Comment 1 (requested changes):
>
> The authors thank the reviewer for pointing that out. We have now formally defined negative transfer in the Introduction on Page 2 of the updated paper.  The text now reads as follows:
>
> “In a general sense, a negative transfer can be defined as a scenario of transfer learning where the target task demonstrates a lower performance due to transfer knowledge from a source task compared to the case where the target task is trained individually. In an MTL setting, both source and target tasks are optimized together, where one or more tasks dominate learning by overpowering others.
>
> We describe negative transfer in MTL as a scenario where a subset of target task(s) report sub-optimal performance compared to their respective performance when trained in the absence of source task(s). The absence of a source task can be defined as a scenario where the target task is trained individually (single task learning: STL) or the target task is trained in an MTL setup without the source task in consideration.”
>
> 2. Comment 2 (requested changes):
>
> We have now formally defined negative transfer in the context of 2+ tasks in the Introduction and in Section 1.1 of Background and Problem Formulation (Page 2 of the updated paper). In the Introduction, the text now reads as follows:
>
> “In the context of MTL, source and target domains are trained together and interchangeably transfer knowledge. With multiple tasks optimized together, a subset of task(s) may dominate training. Here, the performance for these dominant tasks improves while the tasks outside the dominant group observe lower performance (Liu et al., 2019a), referred to as negative transfer in MTL. We describe negative transfer in MTL as a scenario where a subset of target task(s) reports sub-optimal performance compared to their respective performance when trained in the absence of source task(s). The absence of a source task can be defined as a scenario where the target task is trained individually (single task learning: STL) or the target task is trained in an MTL setup without the source task in consideration. This can be seen in Figure 1 under the MTL setup.”
>
> In Section 1.1, the negative transfer is defined mathematically as follows (on Page 3):
>
> “To motivate the problem of negative transfer, let us consider task interference on the shared parameters $\theta^k_{sh}$. During parameter updates for each sample $i$ (or minibatch), the gradients $\triangledown_\theta$ flow backwards from task-specific $\theta^k_{t}$ to the shared parameters $\theta^k_{sh}$. The parameters $\theta^k_{sh}$ are updated by a linear combination of individual task’s gradients, as given by: $\triangledown_\theta^{sh}L = \sum_{t=1}^{K} \triangledown_\theta L_{t_i}$. However, at these shared layers, the gradients of different tasks may interfere with opposite update directions. The disagreement in the gradient directions between the tasks may nullify the overall gradient, limiting performance for a subset of task(s). For instance, as seen for MTL in Figure 1, gradients from $T_A$ nullify gradients for $T_B$ and $T_C$ at the shared layer.”
>
>
> 3. Comment 3 (requested changes):
>
> We have now added sample outputs from all the other comparative algorithms in Figure 7 on Page 19 of the updated paper.

---

### Review · Reviewer_doDg · 2022-10-07

**Summary Of Contributions:**

This paper deals with the negative transfer problem in multi-task learning. The authors tackle the problem from a training tasks schedule perspective and propose the Dropped Scheduled Task (DST) algorithm.

Specifically, for each training task, the drop probability is determined by four metrics (two statics and two dynamics): network depth, ground-truth samples, training incompleteness, and training stagnancy.
The former two are static ones according to the model and dataset statistics, and the latter two are dynamic ones either based on the per-epoch loss values or on the first-order difference between consecutive epoch loss values.
In addition, a constant regularization metric is given.
The final dropping probability is calculated from the weighted sum of all five metrics.

Three sets of experiments are performed to verify the proposed algorithm: unilateral multi-task face applications, multilateral fingerprint applications, and multi-task Omniglot character recognition. Comparison with several baseline methods shows the effectiveness of the proposed DST algorithm. Ablation studies give intuitions about each designed metric in mitigating negative transfer.

Overall, the contribution lies in two aspects: (1) the designed metrics to determine the dropping probability in the multi-task setting, (2) experiments on three multi-task settings show the effectiveness.


**Requested Changes:**

- Recent and most related comparison methods
- Correct the formulations
- Correct other formations and typos

Others, please refer to the above weaknesses.


**Strengths And Weaknesses:**

## Strengths
- The main strength is the comprehensive investigation of the proposed DST algorithm. A series of experiments and analyses are conducted to investigate the DST algorithm.
- The proposed DST algorithm is verified to be effective compared with several reported MTL algorithms. The overall average performance exceeds the baselines.
- Ablation and visualization give some insights into the proposed five metrics and their effect.

## Weaknesses
- From the algorithm and method perspective, the proposed DST is simple and ad-hoc, involving the manually designed metrics at training time or based on training statistics. I am concerned about whether these metrics can be generalized when the test statistics are different from the training.
- Although DST outperforms several baselines, except for Gradient Drop, all others are before 2020. The authors mentioned several other related methods of mitigating negative transfer, e.g., Liu et al. (2022), Sun et al. (2021), and Lee et al. (2018) (earlier one, but also focus on negative transfer). Is it possible to compare any of these new/related methods in the experiment?
- There is a lacking of Omniglot settings and experiment details either in the method, experiment, or appendix section.
- Formulation Equation issues:
(1) Incorrect formulation, Eq. (11), (12), and (18) are incorrect. For example, Eq. (11) should be $T_{1F_i} = -g_i \log (P(g_i|X_{F_i}^{'}))  -(1-g_i) \log (1-P(g_i|X_{F_i}^{'}))$, or alternatively define $g_i \in \mathbb{R}^{2}$ as a 2-dim formation.
(2) No definition for symbols, e.g., before sec 2.1,  $P_{(d,t)}, P_{(c,t)}, P_{(u,k,t)}$, and $P_{(r,k,t)}$ used before defined. Also, it is unclear what are $u$ and $r$ mean in the metrics.
(3) There is no punctuation after each equation.
(4) In Eq. (20), how is $D_i$ implemented?
- Typos:
For example, "easier tasks(Lee et al., 2018)", and " explored(Han et al., 2017; Ranjan et al., 2017)." miss a space before the left parentheses.
- Formation issues:
(1) Figure 2 is not in the proper place;
(2) Unexpected space areas on Pages 11, 13, and 16;
(3) The rotated results table is OK but seems a little uncomfortable for the audience.

I acknowledge the efforts regarding experiments the authors made in this paper, but I think the proposed method is kind of simple and ad-hoc. Also, there remain several issues that show the paper is not ready.

---

> ### Author Response · Authors · 2022-11-17
> **Detailed Response to comments of Reviewer doDg (Part 1: Addressing weaknesses)**
>
> 1. Comment 1 (from weakness): These metrics are there to optimize the training procedure and limit negative transfer. All these metrics exist to optimize training procedures and are backed with optimal results by the proposed DST algorithm across datasets, applications, and comparative algorithms. Testing statistics don’t involve task incompleteness and stagnation, as no training is involved. Similarly, architecture remains the same throughout training and testing. Thus the metric based on network depth would remain the same as well. Lastly, as illustrated in Table 2 in the updated paper (and correspondingly in Section 3.1 Unilateral MTL), the ground truth labeled sample count differs for each task during model training. These varying proportions of labels for training are selected for each task by us to showcase the scenario where labels can be unavailable. Hence, this metric based on training sample count, presented in Section 2.1.2, becomes useful. However, the testing statistics already differ during testing as all testing labels are considered.
>
> 2. Comment 2 (from weakness; additional comparisons): As suggested by the reviewer, we have added these references in the literature review and respective comparative results with Liu et al. (2022), Lee et al. (2018), and Maninis et al. (2019) in Tables 4, 5, and 7 (in the updated paper). Correspondingly, we have also added a review in the literature review, analysis in Section 4, and output illustration in Figure 7 for the newly added comparative results.
>
> 3. Comment 3 (from weakness; details of Omniglot): As suggested by the reviewer, we have elaborated on the Omniglot database and protocol details in Section 3.1 on Page 12 of the updated paper. Additionally, the architecture and results are further elaborated in Section 4.3 (Page 18).
>
> 4. Comment 4 (from weakness; Formulation Equation issues):
>
> (4A.) We have expanded Eq. (11) and Eq. (12) to illustrate binary cross entropy. We have also rectified Eq. (18) for the 6-class classification
>
> (4B.) The symbols P(d,t), P(c,t), P(u,k,t), and P(r,k,t) are now defined at their first occurrence in the second paragraph of Section 2 (on Page 5 of the updated paper). The variable u in P(u,k,t) is an alias for the incompleteness measure, as captured by I(k,t). Similarly, the variable r in P(r,k,t) is an alias for the stagnancy measure, as captured by R’(k,t). We have reflected the above two statements after Eq. (4) and Eq. (7) on Page 7 of the updated paper, respectively.
>
> (4C.) Punctuation marks have now been added after each equation.
>
> (4D.) $D_i$ is implemented as Euclidean distance between the two encoder representations. We have added the word ‘Euclidean’ just before Eq. (20) on Page 11 of the updated paper.
>
> 5. Comment 5 (from weakness; Typos): We have fixed these in the updated paper.
>
> 6. Comment 6 (from weakness; Formation issues): We have addressed each of the sub-comments individually as follows:
>
> (6A.) We have moved Figure 2 closer to Section 2.2, where it is suitable to explain different tasks associated with the application of face MTL. We have fixed the position of other figures as well.
>
> (6B.) We have fixed the formatting issues in the paper and removed any unexpected spaces.
>
> (6C.) We understand the discomfort caused by rotated pages. Hence, we have moved some content from landscape to portrait, including Figure 5, Figure 6, and Table 8. Now, only Tables 4, 5, and 9 remain rotated in landscape mode in the main paper as these require longer horizontal spaces.

---

> ### Author Response · Authors · 2022-11-17
> **Detailed Response to comments of Reviewer doDg (Part 2: Addressing weaknesses and Requested Changes)**
>
> 7. Comment 7 (from weakness; simplicity of proposed method):
> Response: We thank the reviewer for pointing out all the above mentioned concerns in Comments 1 to 6 and other reviewers, too, for their valuable feedback. We have incorporated suggested changes in our updated paper, which we believe has significantly improved our proposed work due to the feedback provided by the reviewers. The proposed research study first defines negative transfer in the context of MTL, and then the proposed DST algorithm builds upon various metrics to counter the effect of negative transfer. Overall, the contributions of the DST algorithm are as follows:
>
> (7A.) The DST algorithm considers negative transfer caused due to task-wise varying network depths and limited samples for some tasks. Due to network depth as a parameter, DST is shown to work in unilateral and bilateral models.
>
> (7B.) The DST algorithm computes the completeness and stagnation of the tasks, allowing only a subset of tasks to remain active. Further, DST allows prolonged stagnated earlier finished tasks to be occasionally active in later training stages, reducing catastrophic forgetting.
>
> (7C.) While few other studies propose prioritizing tasks based on task completeness, we propose to stochastically "drop" tasks that can potentially cause a negative transfer. Due to task dropping, $2^K$ combinations from K tasks are possible. Different combinations during training act as implicit regularization while simultaneously reducing the impact of negative transfer.
>
> (7D.) For experimental results, three applications are chosen related to faces, (latent) fingerprints, and character recognition. The applications are chosen considering: (i) their diverse nature and complexities (classification, image-to-image translation, regression, contrastive loss, segmentation), (ii) different output depths (encoder/FC/decoder) and architecture (unilateral/bilateral), (iii) varying difficulty of similar tasks (segmentation for fingerprints easier than latent fingerprints), and (iv) large number of tasks (Omniglot).
>
> (7E.) While this research focuses on the metrics defined in the DST algorithm for vanilla MTL, we assert that these metrics can also have broader implications and use cases due to its straightforward implementation. For instance, task incompleteness and task stagnation metrics can be extended for the case of single-task classification models, where the aim is to increase the performance of a weaker class. Similarly, the metric on task depth and training sample count can become vital in MTL with auxiliary tasks. In such a scenario, these metrics can be used as it is or altered to prioritize the main task more compared to other auxiliary tasks.
>
> ADDRESSING REQUESTED CHANGES
>
> 1. Comment 1 (from requested changes; additional comparison): We have added results and related comparisons with three more recent papers, as suggested by the reviewer in Comment 2. Accordingly, we added analysis and output illustrations in Section 4 and Figure 7, respectively.
>
> 2. Comment 2 (from requested changes; formulations): We have corrected the formulations. More details in response for Comment 4 of the paper's weaknesses.
>
> 3. Comment 3 (from requested changes; formations and typos): We have fixed the formatting and typos as suggested by the reviewer in Comments 5 and 6 of the paper's weaknesses.

---

### Review · Reviewer_psS9 · 2022-11-02

**Summary Of Contributions:**

This paper proposes a multi-task learning framework, which differs from previous work in that the authors define several metrics for "dropping" the task in the multi-task training framework so that the tasks are dynamically dropped. The authors claim that this dropped scheduled task can help remove the negative transfer from dominant tasks. The four metrics are (1) task depth, (2) number of ground-truth samples per task, (3) amount of training completed, (4) task stagnancy. Based on these metrics, a scheduling probability is decided then the tasks can be dropped during training. Then the authors conduct experiments on three types of applications with multiple tasks. The results show that the method is effective compared to previous works.

**Requested Changes:**

See above.

**Strengths And Weaknesses:**

Strengths:
1. The auhtors propose a different way for multi-task training. The novel point is from the task "dropping", which is different from previous works. In this drop way, the tasks can be dynamically decided to be training or not.
2. The authors define different metrics for deciding the probability of the task drop, and these metrics are reasonably defined by considering the different aspects of the task and the training status.
3. The experiments show effectiveness in some applications.

Weaknesses:
1. Though the four metrics are reasonably defined by considering the different views. One feeling is that these metrics are somehow heuristic and straightforward. It's hard to see the technique's novelty in terms of these metrics.
2. One big concern of the paper is the definition of "negative transfer". While the authors claim the negative transfer needs to be avoided and this is the main motivation for this method. It is important to mathematically define what "negative transfer" is. The best way is with some statistical measurement.
3. In terms of writing, there are some typos that need to be revised. For example, "Furthermore, The xxx". Besides, $k$ and $K$ represents differently. It is highly suggested to use unified notation. For example, $k$ and $K$ for layer, and $t$ and $T$ for tasks.
4. It is also noted in Table 3, the results of multi-task training are not good as individual training. Therefore, this is not to show the effectiveness of multi-task training. Besides, compared to some previous works, the improvements are not consistent. This is a serious problem, but more modifications of the method are encouraged to achieve better performance. Figure 5 is not so clear, for example, the legend in the figure, and the comparison of the figure is not so obvious. It can be removed then.

---

> ### Author Response · Authors · 2022-11-17
> **Detailed response to comments of Reviewer psS9**
>
> 1. For designing the proposed DST algorithm, we discovered 4 potential gaps that adversely affect an MTL network. These 4 metrics are intentionally defined to be simple for a wider application. For instance, task incompleteness and task stagnation metrics can easily be extended for the case of single-task classification models, where the aim is to increase performance for a weaker class. Similarly, the metric on task depth & training sample count can become vital in MTL with auxiliary tasks. In such a scenario, these metrics can be used as it is or altered to prioritize the main task more compared to other auxiliary tasks. Furthermore, while the presented metrics may seem straightforward, they are not ad-hoc. The concepts of vanishing gradients with increasing depths, negative transfer due to imbalance in sample counts, and task prioritization are also validated by other researchers. For example, Wu et al. [1] showed in their experiments that the negative transfer reduces and eventually shifts to positive transfer when sample count is increased in the source task (in MTL). This lays the foundation of the metric of the task’s sample count. Similarly, researchers also highlight the issue of vanishing gradients in MTL, eventually addressed with additional loss term [2] or residual connections. Also, utilizing task’s starting loss or validation performance is widely used for task prioritization. Lastly, a few strategies have also been explored to quantify descending rates [3].
>
> [1] Wu et al., Understanding and Improving Information Transfer in MTL. ICLR. 2020.
>
> [2] Qiu et al., Adversarial MTL with inverse mapping for speech enhancement. Applied Soft Computing, 2022.
>
> [3] Liu et al., End-to-end MTL with attention. CVPR. 2019.
>
> 2. In the newly added Section 1.1 of Background and Problem Definition, we have mathematically defined negative transfer (Pages 2-3). Also, we statistically quantify the impact of the negative transfer on vanilla MTL with the help of Table 1. In Table 1, we see there is a limited correlation in task gradients, due to which tasks may nullify each other during joint optimization. This supports the impact of negative transfer caused by interfering gradients in MTL.
>
> 3. We have fixed the typos in the updated paper. We used the variables T, t, and K to discuss different aspects relating to tasks. To elaborate them, T is used to explain different tasks in the considered applications. The variable t refers to a single task in an MTL setup and K showcases the total number of tasks. Additionally, we agree that the paper requires clarity on notations. Hence, we have added Section 1.1 to define the problem & its corresponding notations.
>
> 4. First, we address the sub-comment that DST is not good as individual training (STL) in Table 4 of the updated paper. To address that, we discuss the errors reported in Table 4 for STL vs. MTL vs. DST (lower the better). The proposed study reduces negative transfer in MTL by recovering the loss in overall performance. As seen in Table 4, the avg. error for STL is 10.87%, which is sub-par with the performance of vanilla MTL, which has an avg. error of 11.29%. The easier task of gender recognition (T1) overpowers joint optimization. Hence, the performances of T3, T4, and T5 suffer. The proposed DST reduces negative transfer and improves performance with an avg. error of 10.75%. The proposed DST achieves this by limiting the computation cycles for T1 and prioritizing T3 and T5, significantly improving overall performance with gains in T2 to T5.  Secondly, the reviewer mentions that the improvement is inconsistent compared to previous works. For the six columns presented as results (Tasks 1 to 5 and overall avg. error), the proposed DST yields the best performance for four (T3, T4, T5, & overall) and the second-best performance for T2 across all MTL algorithms. As mentioned above, the proposed DST achieves the best performance by limiting the computation cycles for T1 and prioritizing T3 and T5. This significantly improves the overall performance with gains in T2 to T5 and slightly sub-par performance for T1. Hence, having lower performance in T1 is justified for the greater common good. The proposed DST algorithm and other comparative algorithms in Table 4 aim to prioritize a subset of task(s) in MTL as per respective logic. While we introduce 4 metrics to schedule tasks, few papers use individual task's loss only. Some algorithms utilize the performance on the train/validation set, while others use gradient values. The way various studies select tasks for prioritization differs; hence, performances have variabilities.  For the third sub-comment on Fig. 5, we have now added the legend. The description to interpret the t-SNE plot for comparison for MTL vs. DST is provided in the paragraph below Fig. 5. We have rewritten parts of the text for better interpretation and briefly elaborated in the caption of Fig. 5. These changes can be seen on Pages 13-14 of the paper.

---

### Decision · Action_Editors · 2022-12-03

**Recommendation:** Accept with minor revision

**Comment:**

This paper proposes a multi-task learning framework, in which the authors define several metrics for "dropping" the task in the multi-task training framework so that the tasks are dynamically dropped. While the specific technical design is a little bit heuristic (multiple reviewers raised their concerns that the technical novelty and depth of the paper is limited, and its implication to more general tasks and its value to general audience are questionable), this is not the main evaluation criteria of TMLR. In terms of the two key criteria (claims and evidence, audience), the authors have done a reasonably good job. Therefore, I believe this paper is acceptable, provided that the authors can make necessary changes according to the detailed review comments.

**Audience:**

Yes

**Claims And Evidence:**

Yes